# Anterior cingulate cross-hemispheric inhibition via the claustrum resolves painful sensory conflict
Keisuke Koga [1] ✉, Kenta Kobayashi [2] ✉, Makoto Tsuda [3], Anthony E. Pickering [4] &
Hidemasa Furue [1] ✉

The anterior cingulate cortex (ACC) responds to noxious and innocuous sensory inputs, and integrates them to coordinate appropriate behavioral reactions. However, the role of the projections of ACC neurons to subcortical areas and their influence on sensory processing are not fully investigated. Here, we identified that ACC neurons projecting to the contralateral claustrum (ACC$^{\rightarrow contraCLA}$) preferentially respond to contralateral mechanical sensory stimulation. These sensory responses were enhanced during attending behavior. Optogenetic activation of ACC$^{\rightarrow contraCLA}$ neurons silenced pyramidal neurons in the contralateral ACC by recruiting local circuit fast-spiking interneuron activation via an excitatory relay in the CLA. This circuit activation suppressed withdrawal behavior to mechanical stimuli ipsilateral to the ACC$^{\rightarrow contraCLA}$ neurons. Chemogenetic silencing showed that the cross-hemispheric circuit has an important role in the suppression of contralateral nociceptive behavior during sensory-driven attending behavior. Our findings identify a cross-hemispheric cortical-subcortical-cortical arc allowing the brain to give attentional priority to competing innocuous and noxious inputs.

The ACC is important for both sensory integration and the processing of sensory and affective aspects of pain[1], and synaptic plasticity in the ACC is important in the induction of chronic pain[2,3]. Recent studies using genetic tools identified roles for neuronal circuits of the ACC in pain regulation[4–9]. Specifically, optogenetic or chemogenetic activation of ACC pyramidal neurons causes pain hypersensitivity, and conversely, their inhibition alleviates pain hypersensitivity and aversion in rodent pain models[4,10,11]. More recently, a cross-hemispheric ACC projection has been described that facilitates contralateral ACC neuronal activity and is involved in spatial generalization of pain with bilateral allodynia after unilateral peripheral nerve injury[12]. However, the physiological roles of contralateral projections of ACC to other brain regions, in particular, those mediating inhibitory interactions between the ACCs, have not been characterized.

It is reported that the ACC connects more strongly to the contralateral than the ipsilateral claustrum (CLA)[13]. The CLA has widespread connections mainly with cortical areas and is believed to be an important negative regulator of cortical activity for a range of functions, including multisensory integration, salience detection, sleep state, and unconsciousness[14–16]. The CLA is also important in the cognitive control of action[17–19] and supports behavioral output by suppressing irrelevant sensory inputs and promoting focus[20]. The CLA is known to have reciprocal projections with the ACC[20], but the role of these projections in pain regulation or sensory prioritization is largely unknown[21].

Using viral tracing and in vivo and slice electrophysiology, we show that ACC neurons projecting to the contralateral CLA (ACC$^{\rightarrow contraCLA}$) preferentially respond to contralateral mechanical sensory stimulation, and these responses are enhanced during sensory-driven attending behavior. Opto- and chemogenetic activation of ACC$^{\rightarrow contraCLA}$ neurons induced silencing of the contralateral ACC via the CLA. This circuit was important for contralateral inhibition of mechanical sensation during lateralized attending behavior to the opposite side. Our findings identify a cross-hemispheric mechanism via a cortical–subcortical–cortical arc allowing the brain to give attentional priority to competing innocuous and noxious inputs.

[1]Department of Neurophysiology, Hyogo Medical University, Nishinomiya, Japan. [2]Section of Viral Vector Development, National Institute for Physiological Sciences, Okazaki, Japan. [3]Department of Molecular and System Pharmacology, Graduate School of Pharmaceutical Sciences, Kyushu University, Fukuoka, Japan. [4]Anesthesia, Pain and Critical Care Research, School of Physiology, Pharmacology and Neuroscience, University of Bristol, Bristol, UK. ✉e-mail: ke-koga@hyo-med.ac.jp; hi-furue@hyo-med.ac.jp

## Results

### ACC neurons preferentially innervate the contralateral CLA

To investigate the axonal trajectory of ACC neurons, we expressed a channelrhodopsin variant, Chronos[22], fused with GFP in ACC neurons by microinjection of AAV-hSyn-Chronos-GFP in the left ACC (Fig.1a). The GFP-expressing axonal terminals were distributed broadly in cortical and subcortical areas, e.g., contralateral ACC[12], retrosplenial cortex[23], striatum[24], basolateral amygdala (BLA)[7], mediodorsal thalamus (MD), zona incerta (ZI) and CLA (Fig. 1b). The fluorescent intensity of the GFP-expressing terminals in the CLA was stronger in the right (contralateral to the AAV injection side) than in the left side (Fig. 1c, d), in consistent with a previous report[13], while in the other subcortical regions (MD, BLA ZI and parafascicular nucleus (PaF)) the GFP-expressing terminals were always denser in the left (ipsilateral) hemisphere (Fig. 1d). We further confirmed the lateralized projection pattern to the CLA by injecting an AAV virus containing a pyramidal neuron-selective promoter, AAV5-CaMKIIα-hM4Di-mCherry, into the left ACC (Supplementary Fig. 1a, b), and found that the mCherry-expressing ACC terminals were denser in the contralateral CLA than in the ipsilateral CLA (Supplementary Fig. 1c, d). To explore this asymmetry of ACC neuronal projections, we injected a retrograde tracer (Retrobeads) unilaterally into the right CLA (Fig. 1e) and compared the number of bead-labeled (bead[+]) ACC neurons on both sides. Significantly more bead[+] ACC neurons were retrogradely labeled on the contralateral side than ipsilateral for both layer II and layer III–VI (Fig. 1f, g). To investigate how ACC neurons innervate adjacent areas of the CLA, we injected Retrobeads into the right insular cortex (IC) or dorsolateral striatum (dlStr), and found that the number of bead[+] cells retrogradely labeled

from the IC or the dlStr was extremely lower than that from the contralateral and ipsilateral CLA (Supplementary Fig. 1e–h), suggesting selective innervation to the contralateral CLA from the ACC. These results indicate that a greater number of ACC neurons project to the contralateral rather than the ipsilateral CLA.

### ACC$^{\rightarrow contraCLA}$ neurons preferentially respond to contralateral mechanical stimulation

ACC neurons are known to participate in the processing of noxious and innocuous sensory stimuli[25,26]. To examine the sensory responsiveness of ACC neurons projecting to the contralateral CLA (ACC$^{\rightarrow contraCLA}$ neurons), we injected AAV-hSyn-Chronos-GFP into the left ACC, and the tip of an optical fiber was positioned above the right CLA, allowing optogenetic antidromic identification[27,28] (Fig. 2a, b). The single unit activity of the left ACC neurons in layers III–VI was recorded in awake head-fixed mice. We identified ACC$^{\rightarrow contraCLA}$ neurons by the presence of antidromic responses to light stimulation in the CLA (473 nm, 3 ms) (Fig. 2c). Light stimulation elicited constant latency action potentials (AP) ($9.41 \pm 0.36$ ms; jitter, $0.183 \pm 0.010$ ms, at 20 Hz; $n = 33$). We examined the responses of these identified ACC$^{\rightarrow contraCLA}$ neurons to three mechanical stimuli (0.16 g and 0.6 g von Frey stimulation of the whisker pad and whisker stroking) applied bilaterally. The majority of ACC$^{\rightarrow contraCLA}$ neurons tested (72.7%, 24/33 units) showed evoked APs in response to each of the mechanical stimuli, but the firing rates of the responses elicited by mechanical stimulation applied to the right (contralateral to the recorded ACC neuron) were greater than those elicited by ipsilateral stimulation (Fig. 2d–g).

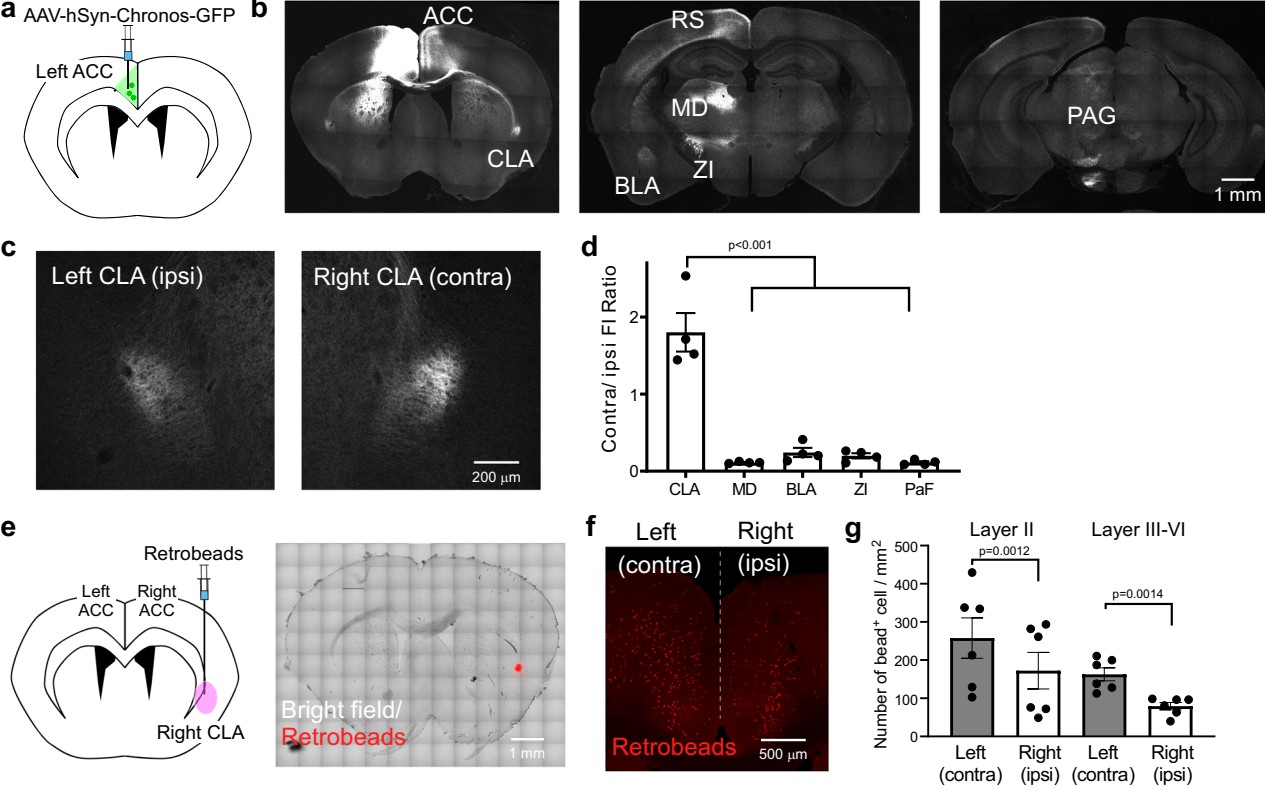

**Fig. 1 | ACC neurons preferentially project to the contralateral claustrum.**
**a** Schematic of AAV (hSyn-Chronos-GFP) vector injection into the left ACC for neuronal circuit mapping. **b** Representative coronal brain sections showing GFP (gray)-expressing axonal terminals of the left ACC neurons. The fluorescence was detected in claustrum (CLA), basolateral amygdala (BLA), mediodorsal thalamus (MD), zona incerta (ZI), periaqueductal gray (PAG), contralateral ACC, striatum, and retrosplenial cortex (RS). **c** Representative images of the ACC neuronal terminals in the left (ipsilateral to the AAV injection side) and right (contralateral) CLA. **d** Quantification of the fluorescent intensity (FI) ratios of the contralateral GFP-

expressing terminals to ipsilateral ones (four mice, one-way repeated measures ANOVA with Dunnett's multiple comparisons test, parafascicular nucleus—PaF). **e** Schematic of retrograde tracing and a representative coronal section showing Retrobeads injection into the right CLA. Retrobeads (red), bright field (gray). **f** A representative coronal section of both sides of the ACC showing Retrobead-labeled (bead[+]) cells. **g** Quantification of the number of bead[+] ACC neurons in the right (ipsilateral to the Retrobeads injection side) and left (contralateral) ACC (6 mice, two-way repeated measures ANOVA with Bonferroni's multiple comparisons test). Error bars show the SEM.

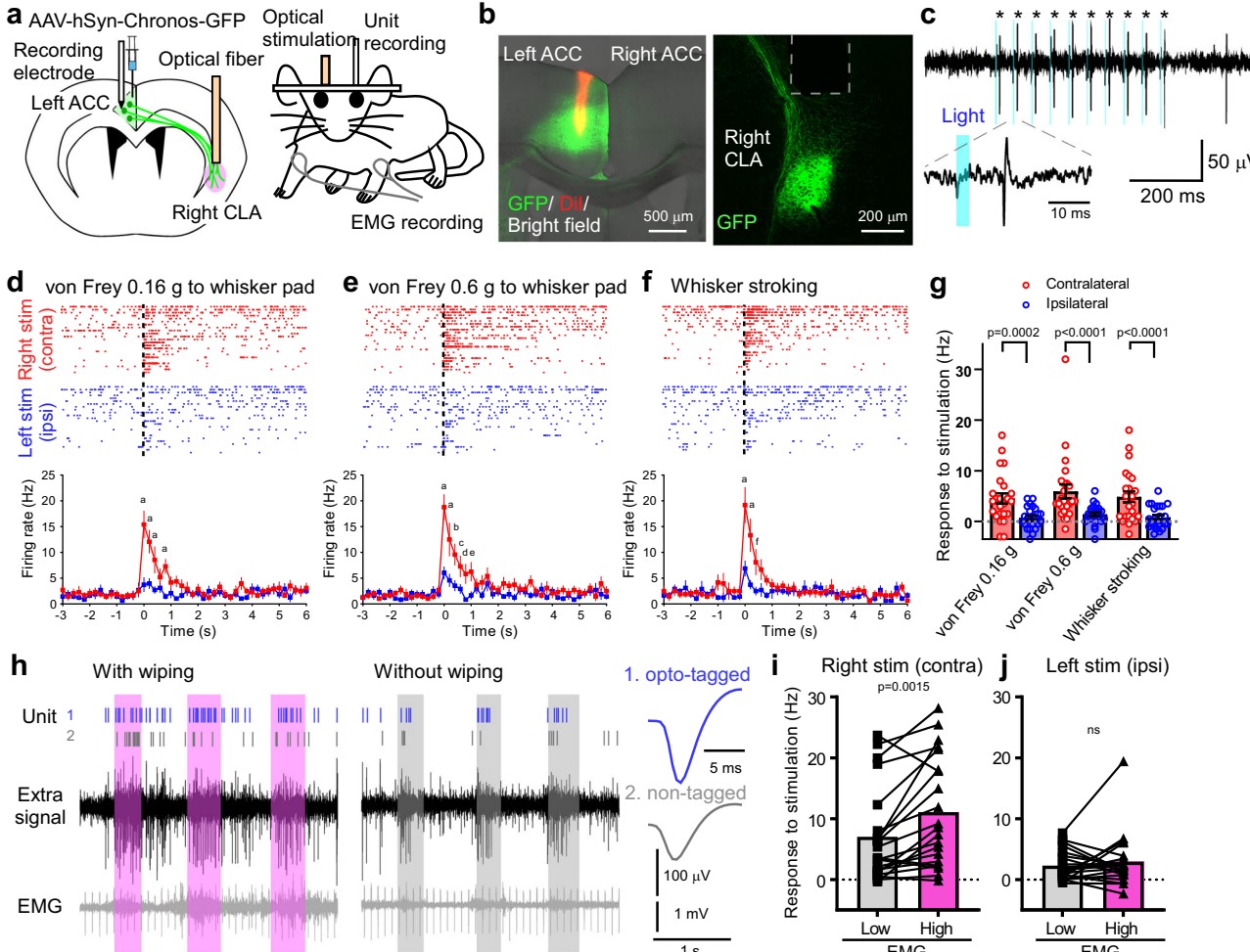

**Fig. 2 | ACC→contraCLA neurons show lateralized responses to mechanical stimulation that are differentially augmented during sensory-evoked attending behavior. a** Schematic of AAV (hSyn-Chronos-GFP) vector injection into the left ACC and unit recording from the left ACC neurons of an awake head-fixed mouse. Optical fiber positioned for antidromic stimulation of the Chronos-GFP-expressing terminals in the right CLA from ACC→contraCLA neurons. **b** Representative images showing the recording site (Chronos-GFP, green; DiI, red) and optical fiber implantation above the CLA (Chronos-GFP, green). **c** Representative traces of extracellular unit recordings showing light stimulation-induced antidromic APs in ACC→contraCLA neurons (indicated by asterisks). **d–f** Single unit activity showing mechanical responses of ACC→contraCLA neurons. Top, response spike raster of ACC→contraCLA neurons in response to von Frey (0.16 and 0.6 g to the whisker pad) and whisker stroking stimulation applied to the right and left whisker pad (contralateral and ipsilateral to the recording side). Each row represents the mechanical response of a different ACC→contraCLA neuron. Dashed lines show the start of the mechanical stimuli. Bottom, the time course of the averaged firing rates of the unit activity of

ACC→contraCLA neurons. The letters indicate the following *p*-values: **a** $p < 0.0001$; **b** $p < 0.001$; **c** $p = 0.031$; **d** $p = 0.0047$; **e** $p = 0.017$; **f** $p = 0.0011$ ($n = 24$ cells, two-way repeated measures ANOVA with Bonferroni's multiple comparisons test). **g** Summary showing firing frequency of mechanical responses of ACC→contraCLA neurons elicited by contralateral and ipsilateral stimulation ($n = 24$ cells, two-way repeated measures ANOVA with Bonferroni's multiple comparisons test). **h** Representative traces of unit activity of ACC→contraCLA neurons in response to whisker stroking stimulation, and simultaneously recorded forelimb EMG. In this recording, two units were isolated from the traces and indicated as 1 (optically tagged) and 2 (non-tagged), and their temporally magnified averaged waveforms and unit activities were shown above each trace. The rhythmic signals in EMG are pulsation signals from the heart. The stimulus timing and duration are indicated in magenta (with attending behavior, wiping) or gray (without wiping). **i, j** Summary showing averaged firing rates in response to whisker stimulation applied to contralateral (**i**) and ipsilateral sides (**j**) with (high EMG) and without (low EMG) wiping behavior ($n = 21$, two-tailed paired *t*-test). Error bars show the SEM.

## ACC→contraCLA sensory responses are enhanced when the mice exhibit sensory-driven attending behavior

The ACC is known to participate in processing multiple aspects of the sensory experience, such as attentional states and affective-motivated behaviors[29]. Following each mechanical stimulus, the mice often exhibited a stereotyped forelimb wipe of their muzzle (using the forelimb on that side, which induced forelimb EMG discharges), a type of attentional behavior[30,31]. Therefore, we compared the responses of ACC→contraCLA neurons elicited by whisker stroking with or without forelimb wiping. The firing rate of ACC→contraCLA neurons in response to the right-sided whisker stroking (contralateral to the recorded neuron) was higher during wiping behavior than on those trials without the behavior (Fig. 2h, i). We further found that no such significant influence of wiping behavior was observed on the

ACC→contraCLA responses to ipsilateral stimulation (Fig. 2j). These observations suggest that ACC→contraACC neurons preferentially respond to contralateral mechanical stimulation, and the responses are enhanced during sensory-driven attending behavior.

## A cross-hemispheric ACC to ACC inhibitory circuit arc relays via the CLA

CLA neurons project ipsilaterally to densely innervate higher-order cortical areas, including the ACC[16]. To examine the synaptic connections of ACC→contraCLA neurons to CLA neurons projecting to the ACC, we injected AAV-hSyn-Chronos-GFP into the left ACC, followed by the injection of Retrobeads into the right ACC 3 weeks after the first AAV injection. Subsequently, whole-cell recordings were made from Retrobead-labeled CLA

(CLA$^{\rightarrow ipsiACC}$) neurons in brain slices (Supplementary Fig. 2a). The Chronos-GFP-expressing terminals of ACC$^{\rightarrow contraCLA}$ neurons in the right CLA were noted to be located in close proximity to the CLA$^{\rightarrow ipsiACC}$ neurons (Supplementary Fig. 2b). Recordings from CLA$^{\rightarrow ipsiACC}$ neurons showed they had a spiny morphology (Supplementary Fig. 2b) and strongly adapting firing patterns including a bursting firing (Supplementary Fig. 2c), as reported previously[17,32]. Optogenetic activation of ACC$^{\rightarrow contraCLA}$ terminals elicited EPSCs in all CLA$^{\rightarrow ipsiACC}$ neurons tested (n = 12/12), and these synaptic responses were inhibited by the AMPA/kainate receptor antagonist, CNQX (control, 247.8 ± 72.4 pA; CNQX, 12.4 ± 2.7 pA; n = 5, $P < 0.05$, two-tailed paired t-test). In the presence of TTX and 4-AP, the light-evoked EPSCs were still detected in 5/6 CLA$^{\rightarrow ipsiACC}$ neurons (Supplementary Fig. 2d, e), indicating that ACC$^{\rightarrow contraCLA}$ neurons make monosynaptic glutamatergic contacts with CLA$^{\rightarrow ipsiACC}$ neurons, inconsistent with previous reports[33,34]. This ACC$^{\rightarrow contraCLA}$-evoked EPSP generated AP discharge in 7/12 CLA$^{\rightarrow ipsiACC}$ neurons (Supplementary Fig. 2f), indicating efficient information transfer from contralateral ACC via the CLA neurons towards the ipsilateral ACC.

We next determined which type of ACC neurons received the synaptic input from the post-synaptic CLA neurons of the contralateral ACC (CLA$_{\leftarrow contraACC}$ neurons). To investigate the innervation of CLA$_{\leftarrow contraACC}$ neurons, we injected an anterograde transsynaptic AAV1 virus[35] into the left ACC to express Flpo recombinase anterogradely from the ACC (AAV1-EF1α-Flpo). This was followed by injection of a Flp-dependent Chronos-GFP expressing viral vector, AAV$_{DJ}$-fDIO-Chronos-GFP, into the right CLA (Fig. 3a). We observed robust expression of Chronos-GFP in anterogradely labeled neurons in the right CLA (CLA$_{\leftarrow contraACC}$). As expected, the GFP-expressing terminals of these CLA neurons were mainly detected in the right (ipsilateral) ACC (Fig. 3b, Supplementary Fig. 3, see also the lower image in Fig. 3d). Recordings from the CLA$_{\leftarrow contraACC}$ neurons in the right CLA confirmed that they expressed sufficient Chronos to be optogenetically driven to fire APs in response to light stimulation (Fig. 3c). To examine the synaptic inputs from CLA$_{\leftarrow contraACC}$ neurons to the ipsilateral ACC, we recorded from layer III–VI neurons. The recorded ACC neurons were categorized into regular-spiking (layer III–VI pyramidal neurons) and fast-spiking interneurons (INs) according to their response to depolarizing current injections (Fig. 3d, e, h). Light stimulation of the Chronos-expressing terminals of the CLA$_{\leftarrow contraACC}$ neurons in the ACC evoked EPSCs in both types of ACC neurons (Fig. 3f, i) (regular spiking, 10/17; fast-spiking, 12/12). We also performed electrophysiological recordings from neurons in superficial layers (Fig. 3k, l, o, p), and they also exhibited light-evoked EPSCs (Fig. 3m, q) (layer II pyramidal, 11/15, layer I INs, 11/13). Although the cell capacitance was larger in pyramidal neurons than fast-spiking INs and layer I INs, suggesting pyramidal neurons have larger cell surface areas (Fig. 3s), the amplitude of light-evoked EPSCs was significantly greater in fast-spiking INs than the other neurons (Fig. 3t). There were no significant differences in the paired-pulse ratio (PPR) of the light-evoked EPSCs (Fig. 3u). Accordingly, the amplitude of light-evoked EPSPs (excitatory post-synaptic potentials) of fast-spiking INs was significantly higher than those of the other neurons (Fig. 3g, j, n, r, v), and these synaptic inputs could only evoke APs in the fast-spiking INs (layer III–VI pyramidal, 0/17; fast-spiking INs, 4/12; layer II pyramidal, 0/15; layer I INs, 0/13). These findings suggest that CLA$_{\leftarrow contraACC}$ neurons make excitatory glutamatergic synaptic contacts ipsilaterally with ACC neurons and preferentially activate fast-spiking INs (inhibitory INs) in the ACC.

### Activation of ACC$^{\rightarrow contraCLA}$ neurons induces a cortical down-state and inhibits sensory responses in the contralateral ACC

To investigate how ACC$^{\rightarrow contraCLA}$ neurons modulate neuronal activity of the downstream target, the contralateral ACC, we recorded local field potential (LFP) and single-unit activity in the deep layers of the right ACC of the awake head-fixed mice in which AAV-hSyn-Chronos-GFP was injected into the left ACC and an optical fiber was positioned above the right CLA (Fig. 4a). Optogenetic stimulation of the terminals of ACC$^{\rightarrow contraCLA}$ neurons

in the right CLA induced a long lasting (100–150 ms) silencing of spontaneous unit activity of pyramidal neurons in the right ACC with large positive slow waves (SWs, 0.5–4 Hz) evident in the LFP (shown in Fig. 4b). This ACC$^{\rightarrow contraCLA}$ optogenetic activation (2 Hz for 2.5 s) significantly increased 0.5–4 Hz EEG delta power (Fig. 4c). Consistently, light stimulation decreased unit activity of putative pyramidal neurons for ~150 ms and transiently increased that of putative interneurons (Fig. 4d–f), as shown previously in the actions of direct claustral stimulation[15]. Next, we tested whether stimulation frequency could affect the EEG delta power increase using different stimulation protocols (2, 5, 10, or 20 Hz, for 2 s), and found that each light stimulation increased the EEG delta power in the right ACC (Supplementary Fig. 4a, b), but the power increase induced by 20 Hz stimulation was significantly lower than those of the other frequencies (Supplementary Fig. 4c), suggesting efficacy of the SW induction depends on ACC input manners. Furthermore, to confirm these effects were mediated by the contralateral CLA, we performed optogenetic inhibition experiments to control the activity of CLA$^{\rightarrow ipsiACC}$ neurons with optical activation of ACC$^{\rightarrow contraCLA}$ neurons. We injected a retrograde AAV expressing Cre recombinase (AAV2-retro-EF1α-Cre)[36] into the right ACC and an AAV expressing eNpHR3.0-EYFP, an inhibitory opsin, in Cre dependent manner (AAV9-EF1α-FLEX-eNpHR3.0-EYFP)[37] into the right CLA, enabling the restricted and functional expression of eNpHR-EYFP in the right CLA$^{\rightarrow ipsiACC}$ neurons (Supplementary Fig. 5a–c). Using the above strategy with expression of Chronos-GFP in the left ACC neurons and optical fiber implantation on the right CLA (Supplementary Fig. 5d), we simultaneously manipulated the ACC$^{\rightarrow contraCLA}$ neuronal terminals by blue light and CLA$^{\rightarrow ipsiACC}$ neurons by red light (633 nm) in the right CLA. Then, we performed LFP recording from the right ACC of the mice and found that inhibition of the right CLA$^{\rightarrow ipsiACC}$ neurons by red light significantly suppressed the EEG delta power increase induced by blue light stimulation of the left ACC$^{\rightarrow contraCLA}$ neurons (Supplementary Fig. 5e–g), confirming that the contralateral CLA is involved in the EEG delta power increase evoked by activation of ACC$^{\rightarrow contraCLA}$ neurons. These findings suggest that activation of ACC$^{\rightarrow contraCLA}$ neurons is sufficient to activate contralateral ACC interneurons via the contralateral CLA, and induce a feedforward inhibition of contralateral pyramidal neurons in awake mice.

We next investigated how light stimulation applied to the right CLA (containing the terminals of left ACC$^{\rightarrow contraCLA}$ neurons) modulates the responses of the right ACC neurons to mechanical stimuli. CLA opto-stimulation (10 Hz for 10 s) reduced the sensory responses in these ACC neurons evoked by mechanical stimulation applied to the contralateral muzzle (Fig. 4g–j). We therefore hypothesized that activation of ACC$^{\rightarrow contraCLA}$ neurons would modulate behavioral mechanical responses through the inhibition of the contralateral ACC. Then, we performed the von Frey test with optogenetic manipulation of ACC$^{\rightarrow contraCLA}$ neurons (Fig. 4k, l) and found that the optogenetic stimulation of the right CLA terminals of the left ACC$^{\rightarrow contraCLA}$ neurons (10 Hz) in the Chronos-expressing group increased the withdrawal threshold of the hindpaw ipsilateral to the left AAV-injected side (contralateral to the CLA terminals), but these changes were not observed in that of the other hindpaw or those of the control group expressing GCaMP6s (Fig. 4m, n). To investigate the involvement of the CLA in these behavioral alterations, we used the anterograde strategy to express Chronos-GFP in CLA$_{\leftarrow contraACC}$ neurons in the right CLA (Fig. 4o and see also Fig. 3a–c). Activation of the CLA$_{\leftarrow contraACC}$ neurons (10 Hz) significantly increased the withdrawal threshold of the hindpaw contralateral to the stimulated CLA in the Chronos-expressing mice, but light stimulation did not affect that of the other hindpaw or those of control animals expressing YFP (Fig. 4p, q). These data confirm the engagement of the contralateral CLA in the behavioral alterations induced by the activation of ACC$^{\rightarrow contraCLA}$ neurons. Furthermore, we performed behavioral experiments using a chemogenetic strategy, we injected a retrograde AAV expressing Flpo recombinase[36,38] into the right CLA (AAV2-retro-EF1α-Flpo) and an AAV expressing hM3Dq[39], excitatory DREADD, in Flp dependent manner into the left ACC (AAV9-EF1α-fDIO-hM3Dq-P2A-mCherry) (Supplementary Fig. 6a). This

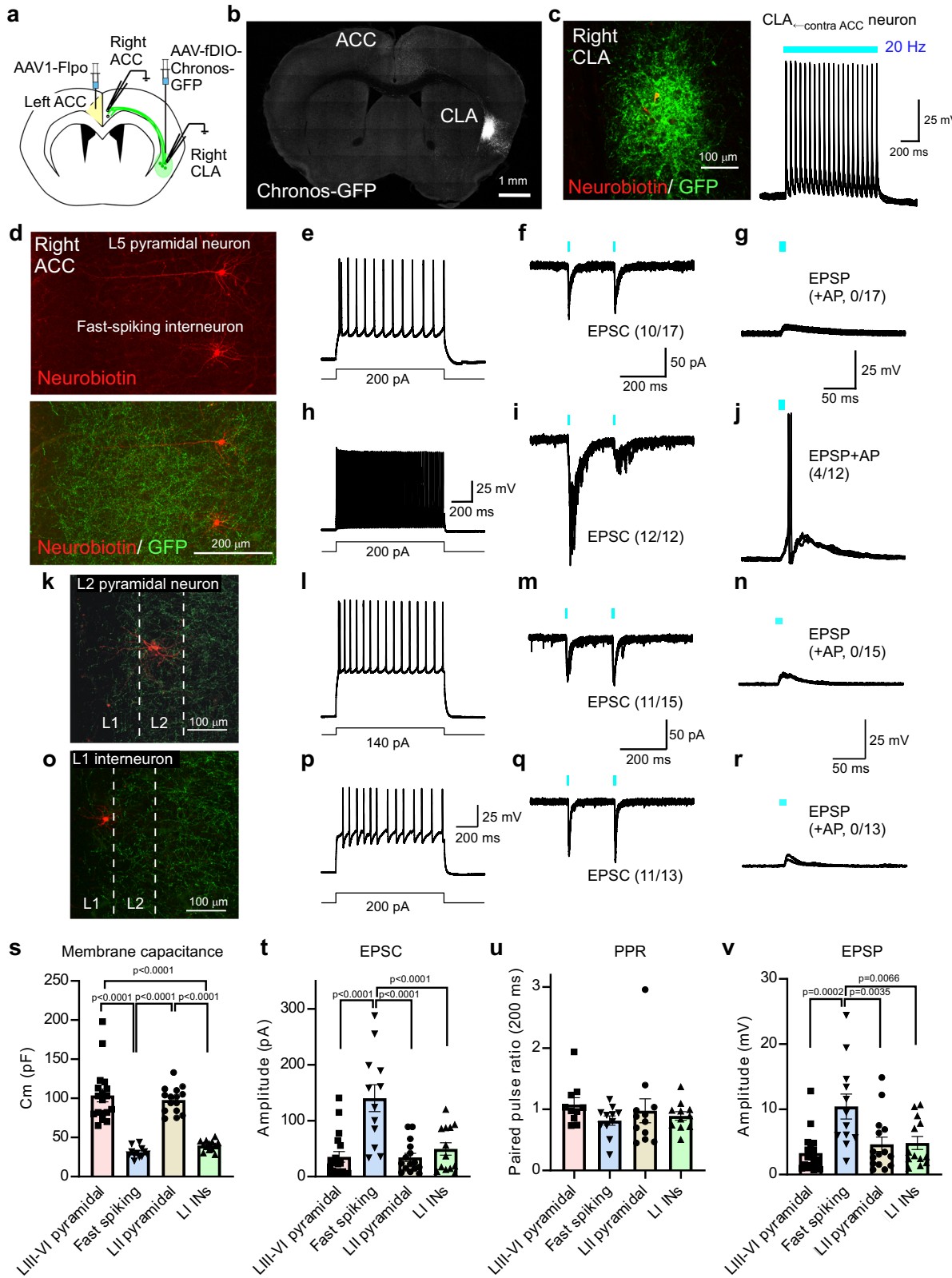

produced hM3Dq-mCherry functional expression in ACC→contraCLA neurons located in layer II–VI of the left ACC (Supplementary Fig. 6b–e). CNO administration (CNO 3 mg/kg, i.p.) to hM3Dq-expressing mice slightly but significantly increased the withdrawal threshold of the paw ipsilateral to the left (hM3Dq-expressing side) ACC compared to control groups (Supplementary Fig. 6g), but not contralateral to the left ACC (Supplementary Fig. 6f). These results suggest that ACC→contraCLA neurons inhibit behavioral responses evoked by mechanical stimulation applied to the ipsilateral paw mediated through the silencing of the contralateral ACC via the contralateral CLA activation.

**Fig. 3 | ACC neurons project to contralateral CLA$^{\rightarrow\text{ipsiACC}}$ neurons that pre-ferentially activate contralateral ACC interneurons. a**–**c** Schematic of anterograde trans-synaptic tracing strategy to express Chronos-GFP in CLA$_{\leftarrow\text{contraACC}}$ neurons (**a**) and a representative image showing the Chronos-GFP-expressing neurons (gray) (**b**). A recorded CLA$_{\leftarrow\text{contraACC}}$ neuron stained with neurobiotin (neurobiotin, red; Chronos-GFP, green) and APs in CLA$_{\leftarrow\text{contraACC}}$ neuron in response to a 20 Hz light stimulation (**c**). **d**–**r** Representative staining images, firing properties, and synaptic responses of pyramidal neurons and interneurons in the right ACC. Representative images of the right ACC showing layer V pyramidal neuron and fast-spiking interneuron stained with neurobiotin (red), and Chronos-GFP-expressing terminals of the right CLA$_{\leftarrow\text{contraACC}}$ neurons (green) (**d**). Representative images of the layer II pyramidal neurons (**k**) and layer I neurons (**o**). Examples of regular spiking firing patterns of the L3–6 pyramidal neuron (**e**), EPSCs (**f**), and EPSPs (**g**) evoked by light

stimulation applied to the Chronos-GFP-expressing terminals. Those of fast-spiking interneurons (**h**–**j**), those of layer II pyramidal neurons (**l**–**n**), and those of layer I interneurons (**p**–**r**). **s**–**v** Summary showing membrane capacitances (**s** L3–5 pyr-amidal, $n = 18$; fast-spiking, $n = 12$; L2 pyramidal, $n = 15$; L1 INs $n = 13$, one-way ANOVA with Tukey's multiple comparisons test), EPSC amplitudes (**t** L3–5 pyr-amidal, $n = 18$; fast-spiking, $n = 12$; L2 pyramidal, $n = 15$; L1 INs $n = 13$, one-way ANOVA with Dunnett's multiple comparisons test), paired-pulse ratios (**u** L3–6 pyramidal, $n = 10$; fast-spiking, $n = 11$; L2 pyramidal, $n = 12$; L1 INs $n = 11$, one-way ANOVA with Dunnett's multiple comparisons test), and EPSP amplitudes (**v** L3–5 pyramidal, $n = 18$; fast-spiking, $n = 12$; L2 pyramidal, $n = 15$; L1 INs $n = 13$, one-way ANOVA with Dunnett's multiple comparisons test), of ACC neurons. Error bars show the SEM.

## ACC$^{\rightarrow\text{contraCLA}}$ neurons are involved in lateralized attentional pain modulation

Because ACC$^{\rightarrow\text{contraCLA}}$ neurons inhibited the behavioral responses elicited by ipsilateral mechanical stimulation (Fig. 4k–n), and their activity in awake mice was enhanced during attending behavior (see Fig. 2), we hypothesized that ACC$^{\rightarrow\text{contraCLA}}$ neurons would suppress withdrawal responses to ipsi-lateral sensory stimulation during contralateral attending behavior.

To induce attending behaviors on the unilateral side, we applied repetitive mechanical stimulation to the unilateral (right) hindpaws of mice (see Methods). The mice exhibited "guarding and lifting" or "licking and biting" of the stimulated (right) hindpaw (Fig. 5a). During these attending behaviors, we applied a von Frey stimulation to the left hindpaw (opposite side to the paw in where attending behaviors were induced). We found that the rate of withdrawal response to the left 1 g von Frey stimulation was increased after the repetitive stimulation when the mice did not exhibit attending behaviors. However, it was strikingly decreased when the mice exhibited "guarding and lifting" or "licking and biting" behaviors (Fig. 5b), suggesting that when the mice exhibit attending behaviors focused on the unilateral hindpaw, their mechanical withdrawal behavior on the con-tralateral side is suppressed.

We examined the same behavioral test using a 2 g von Frey filament as a stronger mechanical stimulation. The withdrawal response rate for the 2 g filament was higher than that for 1 g filament before the repetitive stimu-lation, and it was not altered after the repetitive stimulation during periods without attending behavior (Fig. 5c). However, the withdrawal rate for the 2 g filament was also significantly attenuated during "licking and biting", but not "guarding and lifting" behaviors (Fig. 5c). These results suggest that withdrawal responses to stronger mechanical stimulation are also sup-pressed during a specific attending behavior.

We then examined whether the ACC$^{\rightarrow\text{contraCLA}}$ to contralateral ACC silencing arc was engaged to mediate this phenomenon. After the repetitive right paw stimulation, we observed robust c-Fos expression both in the left and right ACC, but a greater number of c-Fos$^+$ cells were observed in the left ACC (contralateral to the repetitive stimulation) (Fig. 5d; left (contra), $100.1 \pm 17.1$; right (ipsi), $89.7 \pm 15.5$; $n = 5$, $p < 0.05$, two-tailed paired $t$-test). C-Fos expression was also examined in the CLA after repetitive stimulation, and a higher number of c-Fos$^+$ cells were also observed in the right CLA (ipsilateral to repetitive stimulation) (Supplementary Fig. 7). Furthermore, we examined ACC$^{\rightarrow\text{contraCLA}}$ neuronal responses to the repetitive mechanical sti-mulation using fiber photometry method[40], where we retrogradely expressed GCaMP6s[41], a genetically encoded Ca$^{2+}$ indicator, or YFP for control by injection of AAV2-retro-EF1α-Flpo into the right CLA, and of AAV9-EF1α-fDIO-GCaMP6s or AAV9-EF1α-fDIO-YFP into the left ACC (Fig. 5e, f). The GCaMP6s fluorescent signals in ACC$^{\rightarrow\text{contraCLA}}$ neurons on the left side were increased by repetitive mechanical stimulation (for ~20 s) to each hindpaw (Fig. 5g), but the averaged amplitude of response to right (contralateral) stimulation was stronger than that of left (ipsilateral) (Fig. 5h). On the other hand, the fluorescent signals in YFP mice did not change during the repetitive stimulation (Fig. 5g). These findings are consistent with the observations that ACC$^{\rightarrow\text{contraCLA}}$ neurons were preferentially activated by contralateral sensory stimulation to elicit AP discharge in contralateral CLA neurons.

To inhibit neuronal activity in this ACC$^{\rightarrow\text{contraCLA}}$ to contralateral ACC silencing pathway, we retrogradely expressed PSAM$^4$-GlyR[42], an inhibitory chemogenetic actuator, or YFP by injections of AAV2-retro-EF1α-Flpo into the right CLA, and of AAV9-EF1α-fDIO-PSAM$^4$-Gly-P2A-mCherry[43] or AAV9-EF1α-fDIO-YFP into the left ACC (Fig. 5i, j). Whole-cell patch–clamp recording from mCherry-positive ACC neurons showed that varenicline (100 nM, a potent agonist of PSAM$^4$-Gly) significantly inhibited AP firing induced by depolarizing current injections (Fig. 5k, l) as a result of electrical shunting, and also reduced the input resistance (pre, $350.2 \pm 48.9$ MΩ; pre, $165.2 \pm 53.0$ MΩ; $n = 7$, $p < 0.05$, two-tailed paired $t$-test), inconsistent with the previous study[42]. We then examined whether the inhibition of ACC$^{\rightarrow\text{contraCLA}}$ neurons by varenicline affects the reduction of withdrawal responses during attending behaviors. In mice expressing PSAM$^4$-GlyR or YFP in ACC$^{\rightarrow\text{contraCLA}}$ neurons in the left side, varenicline did not affect basal mechanical withdrawal thresholds for either hindpaw (Fig. 5m, n). However, during "guarding and lifting" or "licking and biting" behaviors to the right (contralateral to the PSAM$^4$-GlyR-expressing ACC side) hindpaw induced by repetitive stimulation, the inhibition of the left ACC$^{\rightarrow\text{contraCLA}}$ neurons by varenicline significantly increased the rate of withdrawal responses to 1 and 2 g stimulation applied to the left hindpaw (Fig. 5o). When the mice did not exhibit any such attending behaviors (no nocifensive attending), the response rates were not changed. When attending behaviors were induced in the left (ipsilateral to the PSAM$^4$-GlyR-expressing ACC side), inhibition of the left ACC$^{\rightarrow\text{contraCLA}}$ neurons had no effect on the withdrawal rate of 1 and 2 g stimulation applied to the right (Fig. 5p). To investigate the engagement of the contralateral CLA, a downstream target of ACC$^{\rightarrow\text{contraCLA}}$ neurons, in these behavioral effects, we used an anterograde strategy to express PSAM$^4$-GlyR in the right CLA$_{\leftarrow\text{contraACC}}$ neurons by injecting AAV1-EF1α-Flpo into the left ACC, and AAV9-EF1α-fDIO-PSAM$^4$-Gly-P2A-mCherry into the right CLA (Supplementary Fig. 8a–d). Inhibition of the right CLA$_{\leftarrow\text{contraACC}}$ neurons reproduced the behavioral effects induced by inhibition of the left ACC$^{\rightarrow\text{contraCLA}}$ neurons (Supplementary Fig. 8e–h). Furthermore, these effects were also phenocopied by inhibition of the right ACC inhibitory neurons, downstream targets of the right CLA$_{\leftarrow\text{contraACC}}$ neurons, by CNO using *Vgat-Cre* mice injected with AAV9-EF1α-FLEX-hM4Di-mCherry into the right ACC (Supplementary Fig. 9).

Next, we performed a similar behavioral paradigm by formalin injec-tion to the unilateral (right) hindpaw, which is known to induce sponta-neous nocifensive attending behavior (instead of repetitive mechanical stimulation). After formalin injection (20–30 min), we applied 1 or 2 g von Frey stimulation to the left (opposite side to formalin injection), and found that formalin-induced attending behaviors also inhibited 1 and 2 g von Frey responses of the left hindpaw (Supplementary Fig. 10a–c). This formalin-induced contralateral pain inhibition (right side for formalin injection) was also attenuated by silencing of left ACC$^{\rightarrow\text{contraCLA}}$ neurons, where we used the same strategy as shown in Fig. 5i to express PSAM$^4$-GlyR in the left ACC$^{\rightarrow\text{contraCLA}}$ neurons (Supplementary Fig. 10d). These results suggest that during attending behaviors on one side, behavioral reactions to contralateral mechanical stimulation are inhibited by the engagement of ACC$^{\rightarrow\text{contraCLA}}$ neurons which produce inhibition of the contralateral ACC via the CLA.

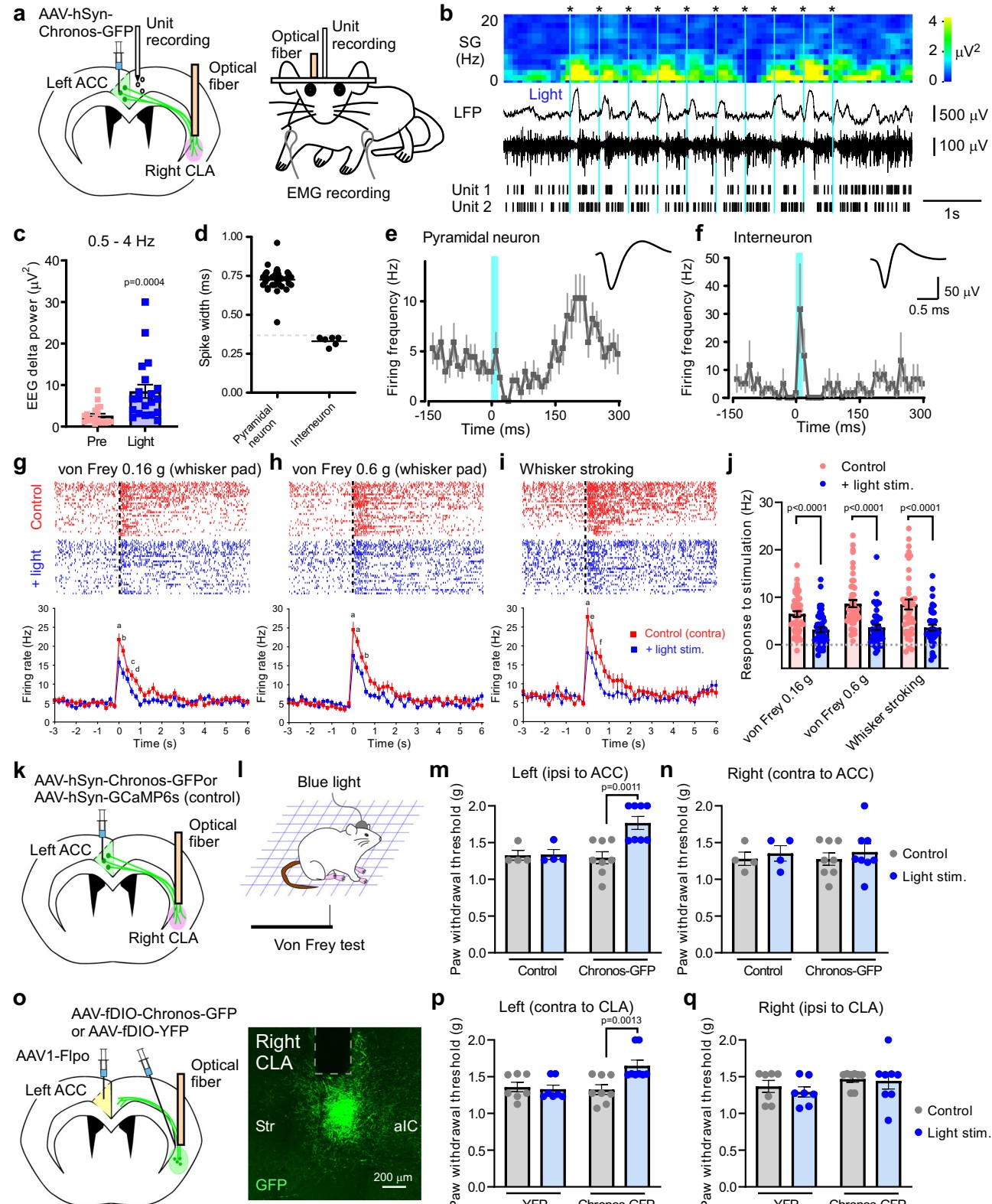

## ACC→contraCLA neurons respond to noxious capsaicin treatment, and this excitation intrinsically inhibits contralateral mechanical hypersensitivity

Because the ACC is implicated in pain hypersensitivity[2], we examined how ACC→contraCLA neurons are involved in capsaicin-induced hyperalgesia. We recorded the unit activity of ACC→contraCLA neurons expressing-Chronos in

the left side (Fig. 6a). Capsaicin injected into the right (contralateral to Chronos-expressing ACC→contraCLA neurons) cheek significantly increased their firing frequency (Fig. 6b). Then, to investigate whether ACC→contraCLA neurons are involved in capsaicin-induced hyperalgesia, we used the same strategy as shown in Fig. 5e to express PSAM[4]-GlyR or YFP in the left ACC→contraCLA neurons (Fig. 6c), and injected capsaicin into the right

**Fig. 4 | Activation of ACC$^{\rightarrow contraCLA}$ neurons induces a cortical down-state in the contralateral ACC and suppresses the somatosensory response. a** Schematic of AAV (Chronos-GFP) injection into the left ACC, and in vivo unit recording from the right ACC of head-fixed awake mice. An optical fiber was placed above the right CLA to activate the Chronos-GFP expressing terminals of ACC$^{\rightarrow contraCLA}$ neurons, and EMG and LFP were simultaneously recorded from the forelimbs and right ACC, respectively. **b** Representative traces of the LFP and extracellular unit recordings. Light stimulation (blue dashed lines, 2 Hz, 10 times) induced SWs and inhibition of unit activity. The spectrogram (SG) and raster display of the two units are shown at the top and bottom of the traces, respectively. **c** Summary of EEG delta power in right ACC before (pre) and during light stimulation (2 Hz, $n = 20$, two-tailed paired $t$-test). **d** Summary showing spike widths of putative pyramidal and interneurons. **e, f** The time course showing actions of the light stimulation (ACC$^{\rightarrow contraCLA}$ neuron terminals, blue line) on firing frequency of pyramidal neurons ($n = 34$) and that of interneurons ($n = 6$). The insets show examples of the unit waveforms. Light stimulation decreased the unit activity of putative pyramidal neurons for ~150 ms followed by rebound firings, and transiently increased that of putative interneurons. **g–i** Spike rasters of representative units (top, each line shows each neuronal spike) and population mean firing rates (bottom) showing mechanical responses in the right ACC evoked by von Frey to whisker pad (**g** 0.16 g, $n = 48$; **h** 0.6 g, $n = 48$) and whisker stroking (**i** $n = 41$) stimulation applied to the left side. The responses without the optical stimulation (control) are shown in red, and those during a repetitive light stimulation (10 Hz for 10 s) were shown in blue. The letters indicate the following $p$-values: **a** $p < 0.0001$; **b** $p < 0.001$; **c** $p = 0.031$; **d** $p = 0.041$; **e** $p = 0.0028$; **f** $p = 0.0016$ (von Frey 0.16 g, $n = 48$; 0.6 g, $n = 48$; whisker stroking, $n = 41$, two-way repeated measures ANOVA with Bonferroni's multiple comparisons test). **j** Summary showing the effect of light stimulation of Chronos-GFP expressing terminals of ACC$^{\rightarrow contraCLA}$ neurons on the mechanical responses in the right ACC neurons (von Frey 0.16 g, $n = 48$; 0.6 g, $n = 48$; whisker stroking, $n = 41$, two-way repeated measures ANOVA with Bonferroni's multiple comparisons test). **k, l** Schematics of AAV (AAV9-hSyn-Chronos-GFP or AAV9-hSy-GCaMP6s (control)) vector injection into the left ACC and optical fiber implantation above the right CLA (k), and behavioral analysis (**l**). **m, n** Summary of the effects of contralateral CLA terminal stimulation of ACC neurons on paw withdrawal threshold of the left paw (**m** ipsilateral to AAV injection, Control $n = 4$; Chronos-GFP = 8, two-way repeated measures ANOVA with Bonferroni's multiple comparisons test) and that of the right paw (**n**, Control $n = 4$; Chronos-GFP $n = 8$). **o** Schematic of anterograde trans-synaptic tracing strategy to express Chronos-GFP or YFP in the right CLA$_{\leftarrow contraACC}$ neurons and optical implantation above the right CLA, and a representative image showing the Chronos-GFP-expressing neurons (green) and the trace of fiber implantation (gray dotted line). **p, q** Summary of the effects of optical stimulation of CLA$_{\leftarrow contraACC}$ neurons on paw withdrawal threshold of the left paw (**p** contralateral to the manipulated right CLA, GFP $n = 7$; Chronos-GFP $n = 8$, two-way repeated measures ANOVA with Bonferroni's multiple comparisons test) and that of the right paw (**q**, GFP $n = 7$; Chronos-GFP $= 8$). Error bars show the SEM.

(contralateral to the PSAM$^4$-GlyR-expressing ACC side) hindpaw. The silencing of the left ACC$^{\rightarrow contraCLA}$ neurons by varenicline did not affect licking time and the subsequent mechanical hypersensitivity for the capsaicin-injected right paw (Fig. 6d, e). However, varenicline induced a mechanical hypersensitivity (lowered threshold) to the left (opposite side to capsaicin injection) paw (Fig. 6f). Furthermore, these behavioral effects were phenocopied by silencing of the right CLA$_{\leftarrow contraACC}$ neurons, downstream targets of the left ACC$^{\rightarrow contraCLA}$ neurons, using the same anterograde strategy as shown in Supplementary Fig. 8a–d to express PSAM$^4$-GlyR in the right CLA$_{\leftarrow contraACC}$ neurons (Supplementary Fig. 8i–k). These findings suggest that ACC$^{\rightarrow contraCLA}$ neurons also respond to capsaicin, and this excitation is intrinsically involved in the inhibition of contralateral mechanical hypersensitivity via the contralateral CLA.

## Discussion

The ACC plays a central role in processing the sensory and affective components of pain to guide behavior, and a number of studies have identified a pain-modulating role for the neuronal circuits of ACC[1,29]. In this study, we showed, via recordings in awake mice, that optogenetically-identified ACC$^{\rightarrow contraCLA}$ neurons preferentially respond to contralateral mechanical sensory stimulation. These responses were enhanced during sensory-driven attending behavior. We show that engagement of ACC$^{\rightarrow contraCLA}$ neurons inhibits the contralateral ACC via the CLA. Chemogenetic manipulations of this cross-hemispheric inhibitory circuit revealed that this circuit mediated suppression of the contralateral withdrawal reflex during sensory-driven attending behavior. These findings define a mechanism by which the ACC influences the processing of spatially distinct (lateralized) and competing sensory input to prioritize and resolve behavioral conflicts (Fig. 7). However, although the projection of ACC neurons to the ipsilateral CLA is weaker than that to the contralateral CLA, further studies are needed to identify the physiological roles of the ACC projection to the ipsilateral CLA.

A recent study reported that ACC neurons with cross-callosal projections to the contralateral ACC are important in the induction of bilateral pain in chronic pain conditions by acting to enhance neuronal activity in both sides of ACC[12]. However, it is unlikely that this pain-facilitating mechanism is the only interaction between the ACC. Therefore, the role of contralateral projections of ACC neurons to subcortical nuclei and their potential inhibitory role in sensory modulation has not been adequately explored. It is known that ACC neurons densely innervate contralateral CLA[34,44], but the sensory responsiveness and behavioral role of ACC$^{\rightarrow contraCLA}$ neurons have not been determined. Mechanical stimulation of whisker pad or whisker of head-fixed awake mice showed that ACC$^{\rightarrow contraCLA}$ neurons preferentially responded to contralateral mechanical stimulation. This activation of ACC$^{\rightarrow contraCLA}$ neurons was stronger when mice exhibited wiping behavioral responses to the mechanical stimuli. This suggests that sensory responses in ACC$^{\rightarrow contraCLA}$ neurons are augmented during attending behavior. A body of literature suggests that ACC neurons in rodents respond to both noxious and innocuous stimuli[45,46], and studies in primates showed that ACC neurons responded to multimodal sensory inputs, including tactile stimulation triggering task movement[47], and this activity represents not only pain itself but also attention to pain and escape behavior[48]. Our findings together with these previous studies, emphasize that ACC activity represents multiple aspects of pain and associated action planning as part of sensory-triggered affective-motivational behaviors. However, the underlying circuit mechanisms controlling the ACC activity levels and behavioral consequences are still unknown, and further studies are needed to reveal them.

The CLA is widely connected with the neocortex and negatively modulates its activity[14,15], and is important for attention by inhibiting cortical representations of distractor sensations[20]. The ACC connects more strongly to the contralateral than the ipsilateral CLA, but the physiological role of this circuit has not been identified. In our study, ACC$^{\rightarrow contraCLA}$ neurons were strongly modulated by sensory-driven attending behavior, activated contralateral CLA projection neurons resulting in activation of fast-spiking neurons, and inhibited sensory responses in the contralateral ACC. The CLA projects predominantly to forebrain regions but also projects more broadly to cortex[16]. We observed that the CLA$_{\leftarrow contraACC}$ neurons mainly project to the ipsilateral cingulate area, but we cannot rule out the possibility that the CLA$_{\leftarrow contraACC}$ neurons suppress pain responses by inhibition of the other sensory-related regions, e.g., S1, S2, IC, and MCC. However, it is reported that optogenetic CLA stimulation induces SW activity most strongly in the frontal cortex reflecting the CLA projection pattern[16]. Furthermore, we show that activation of the ACC terminals in the contralateral CLA strongly induced a SW-like inhibition of spontaneous activity and inhibition of sensory responses in the contralateral ACC, suggesting the contralateral ACC is one of the main targets of CLA$_{\leftarrow contraACC}$ neurons, in accordance with previous studies[16,20].

It is also known that internal states can strongly modulate pain perception by endogenous pain circuit activation[29,49–51]. In the von Frey test paradigm, generally, we measured the paw withdrawal threshold of the mice without attending or affective-motivated behaviors such as exploration or grooming behavior[52]. However, it is not fully characterized how attentional or affective factors modulate pain behavior. We assessed the withdrawal responses of the mice when they exhibited attending behaviors to the

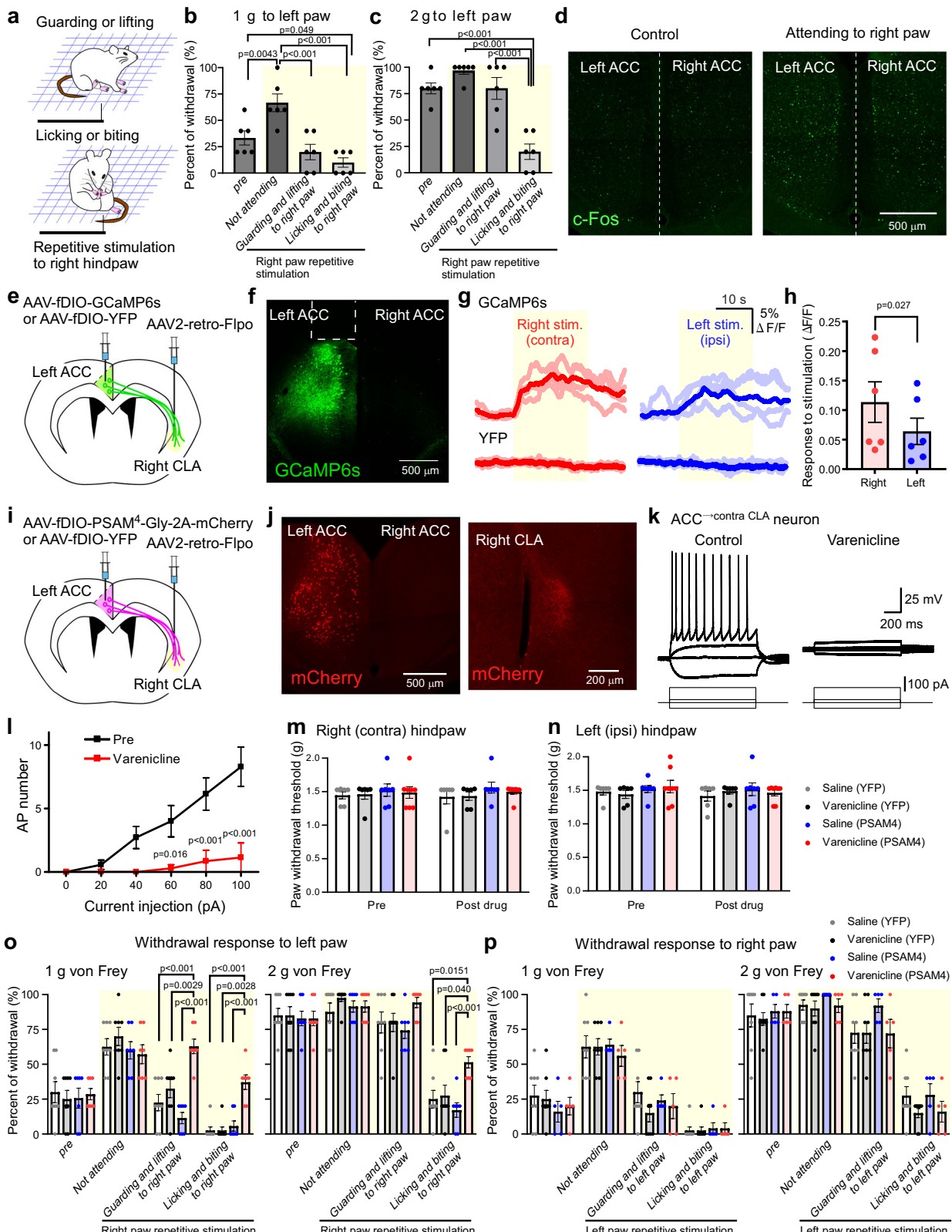

unilateral hindpaw, and showed that the withdrawal responses in the contralateral paw were inhibited. The ACC$^{\rightarrow contraCLA}$ to contralateral ACC inhibitory pathway had an important role in this modulation of contralateral pain responses. Interestingly, in this paradigm, while licking or biting states inhibited both 1 g and 2 g von Frey responses, guarding or lifting states inhibited only 1 g von Frey responses. These results suggest that

the response inhibition was dependent on both the strength of the motivated behavior and the intensity of the sensory "test" stimulation. Similarly, ACC$^{\rightarrow contraCLA}$ neurons responded to capsaicin stimulation, and this was causally shown to be responsible for increasing the mechanical withdrawal threshold on the opposite paw i.e., produced analgesia. These findings are in many ways similar to the conditioned pain modulation paradigm[53] or DNIC

**Fig. 5 | ACC$^{\rightarrow contraCLA}$ neurons are important for attentional contralateral pain modulation. a** Mouse attending to the right hindpaw induced by a repetitive von Frey stimulation. **b, c** Withdrawal rates of the left hindpaw to von Frey stimulations before the repetitive stimulation (pre) and during periods without attending behavior (not attending) and with attending behaviors ("guarding and lifting" or "licking and biting") to the right hindpaw after the repetitive stimulation (**b** 1 g von Frey; **c** 2 g von Frey; $n = 6$, one-way repeated measures ANOVA with Tukey's multiple comparisons test). **d** Representative images of c-Fos expression in the ACC of mice in control (without stimulation) and of mice exhibited attending behaviors to the right paw following the repetitive stimulation (c-Fos, green). **e** Schematic of retrograde transduction strategy of GCaMP6s expression in ACC$^{\rightarrow contraCLA}$ neurons. **f** A representative image of GCaMP6s expression in the left ACC (GCaMP6s, green). The position of fiber implantation was indicated by the dotted line. g Example averaged (thick color) and original (pale color) traces showing fluorescent signals elicited by repetitive mechanical stimulation (for ~20 s, yellow) to each hindpaw (*left traces*, right paw (contra) stimulation; *right traces*, left paw (ipsi) stimulation) obtained from single GCaMP6s-expressing and YFP-expressing mice. **h** Summary showing fluorescent changes during the repetitive stimulation to each hindpaw ($n = 6$, two-tailed paired *t*-test). **i** Schematic of retrograde transduction strategy of PSAM$^4$-GlyR expression in ACC$^{\rightarrow contraCLA}$ neurons. **j** Representative images of mCherry (co-

expressed with PSAM$^4$-GlyR)-expressing neurons in the left ACC and their terminals in the right CLA (mCherry, red). **k** Example traces showing effect of varenicline (100 nM) on APs in response to current injections applied through the recording electrode in mCherry-expressing neurons. **l** Quantification of the effect of varenicline on relationship AP firing number and the amplitude of current injections in mCherry-expressing neurons ($n = 7$, two-way repeated measures ANOVA with Bonferroni's multiple comparisons test). **m, n** Effect of varenicline administration on the paw withdrawal threshold of the right (**m**) and left (**n**) (saline (YFP), $n = 6$; varenicline (YFP), $n = 6$; saline (PSAM$^4$-GlyR), $n = 7$; varenicline (PSAM$^4$-GlyR), $n = 8$, contralateral and ipsilateral to the PSAM$^4$-GlyR- or YFP-expressing ACC side). **o, p** Withdrawal rates of the left (ipsilateral to the PSAM$^4$-GlyR-expressing ACC side) hindpaw to von Frey stimulations in control and during attending behaviors to the right hindpaw (**o** saline (YFP), $n = 8$; varenicline (YFP), $n = 8$; saline (PSAM$^4$-GlyR), $n = 7$; varenicline (PSAM$^4$-GlyR), $n = 7$, two-way repeated measures ANOVA with Bonferroni's multiple comparisons test compared with varenicline (PSAM$^4$-GlyR) group) and of the right hindpaw to von Frey stimulations in control and during left hindpaw attending behavior (**p** saline (YFP), $n = 8$; varenicline (YFP), $n = 8$; saline (PSAM$^4$-GlyR), $n = 5$; varenicline (PSAM$^4$-GlyR), $n = 5$, two-way repeated measures ANOVA with Bonferroni's multiple comparisons test). Error bars show the SEM.

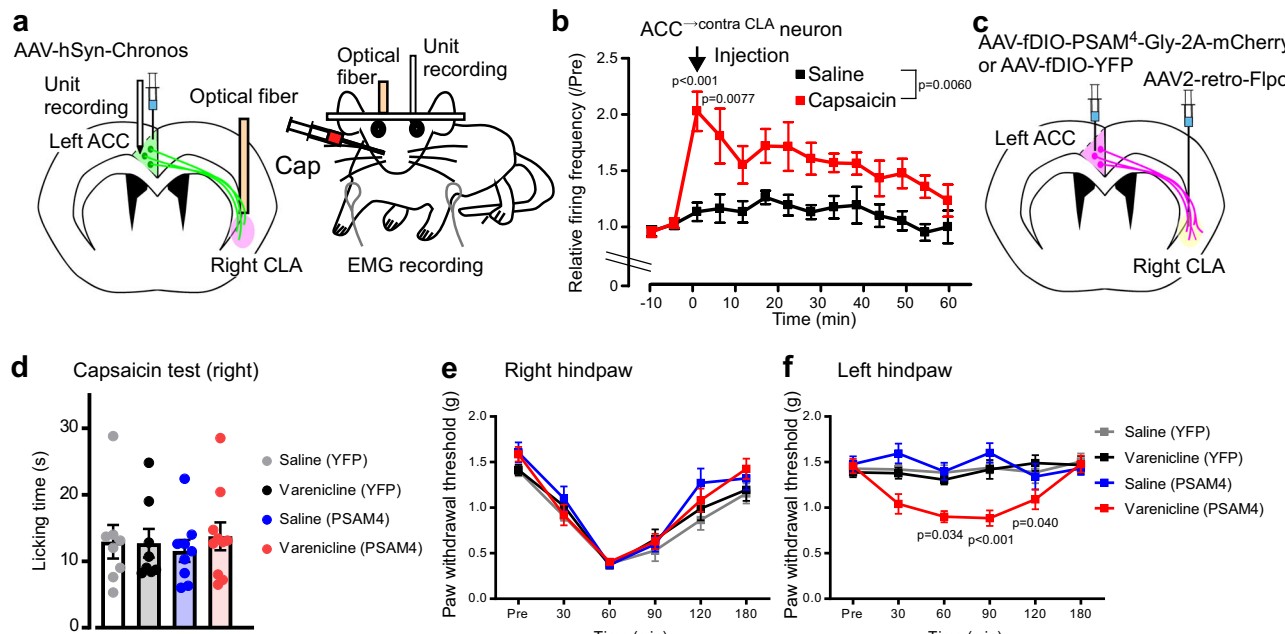

**Fig. 6 | ACC$^{\rightarrow contraCLA}$ neurons are activated by capsaicin stimulation and intrinsically inhibit contralateral mechanical hypersensitivity. a** Schematic of AAV (Chronos-GFP) injection into the left ACC, and in vivo unit recording from the left ACC of head-fixed awake mice. An optical fiber was placed above the right CLA to activate the Chronos-GFP expressing terminals of ACC$^{\rightarrow contraCLA}$ neurons, and capsaicin was injected into the right cheek. **b** Time course of the relative firing frequency of unit activities of ACC$^{\rightarrow contraCLA}$ neurons showing the action of capsaicin and saline injections (saline, $n = 6$; capsaicin, $n = 7$, two-way repeated measures ANOVA with Bonferroni's multiple comparisons test). **c** Schematic of retrograde transduction strategy of PSAM$^4$-GlyR expression in ACC$^{\rightarrow contraCLA}$ neurons of the left

ACC. **d–f** Effect of varenicline administration on capsaicin-induced nocifensive behaviors (licking and biting time) (**d** saline (YFP), $n = 8$; varenicline (YFP), $n = 8$; saline (PSAM$^4$-GlyR), $n = 9$; varenicline (PSAM$^4$-GlyR), $n = 10$, one-way ANOVA with Bonferroni's multiple comparisons test) and on capsaicin-induced mechanical hypersensitivity in the right paw (**e** capsaicin injected side, contralateral to the PSAM$^4$-GlyR-expressing ACC side) and mechanical threshold in the left paw (**f** opposite side to capsaicin injection) (saline (YFP), $n = 8$; varenicline (YFP), $n = 8$; saline (PSAM$^4$-GlyR), $n = 9$; varenicline (PSAM$^4$-GlyR), $n = 10$, two-way repeated measures ANOVA with Bonferroni's multiple comparisons test vs varenicline (YFP) group). Error bars show the SEM.

but without the absolute need for a noxious conditioning stimulus and require the involvement of cortical control rather than just brainstem-level circuitry.

The ACC is involved in assigning salience to sensory events through prioritization and acts as a sensory hub to determine appropriate behavioral strategy[23,29], but the mechanisms of affective evaluation of pain is not fully understood. This study revealed the circuit mechanisms to resolve conflict between noxious and innocuous sensations by prioritizing lateralized perception. Our findings provide clues to understand why pain hypersensitivity is generally limited to the inflamed or injured side i.e., how pain

lateralization is coded in the brain, while the brainstem pain-controlling descending neurons project to both sides of the spinal cord[54]. Our study is critical for understanding how both hemispheres interact and coordinate to determine behavioral consequences for lateralized and/or conflicting sensory information.

## Methods
### Animals
Male C57BL/6 J mice (CLEA Japan) and *Vgat-Cre* mice (B6J-*Slc32a1$^{tm2(cre)}$ $^{lowl}$*/MwarJ, Stock No: 028862, The Jackson Laboratory)[55] were used. All mice

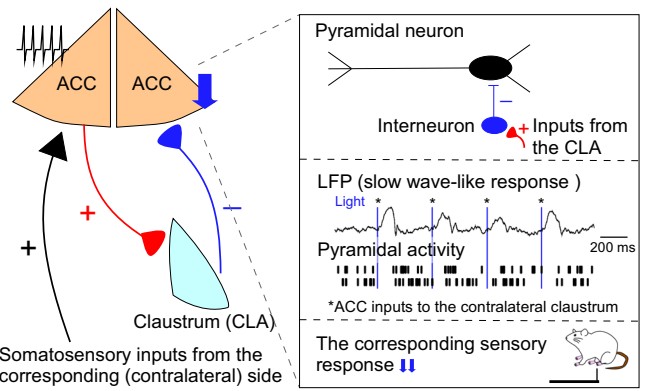

**Fig. 7 | ACC→contraCLA neurons are engaged in the inhibition of the contralateral ACC activity by recruiting local inhibitory circuits via the contralateral CLA and suppressing the ipsilateral mechanical responses.** ACC→contraCLA neurons preferentially respond to contralateral mechanical sensory stimulation. Activation of ACC→contraCLA neurons inhibits the contralateral ACC pyramidal neurons via the CLA by activating local inhibitory circuit and induces EEG delta power increase in the contralateral ACC. This cross-hemispheric inhibitory circuit mediates suppression of the contralateral withdrawal reflex during sensory-driven attending behavior, suggesting that the ACC influences the processing of spatially distinct (lateralized) and competing sensory input to prioritize and resolve behavioral conflicts.

used were 8–12 weeks old at the start of each experiment and were housed at 22 ± 1 °C with a 12-h light–dark cycle with food and water ad libitum. All animal studies were reviewed and approved by the Institutional Animal Care and Use Committee of Hyogo Medical University and were performed in accordance with the institutional guidelines for animal experiments and were consistent with the ethical guidelines of the International Association for the Study of Pain.

## Adeno-associated virus (AAV) production and purification
The pAAV-hSyn-Chronos-GFP (#59170)[22], and pAAV-EF1α-FLPX-rc-[Chronos-GFP] (#122102)[22], pAAV-EF1α-DIO-hM4D(Gi)-mCherry (#50461)[56], pAAV-EF1α-fDIO-YFP (#55641)[36], and pAAV-CaMKIIα-hM4D(Gi)-mCherry (#50477)[56] were purchased from Addgene. AAV9-hSyn-Chronos-GFP, AAV_DJ-EF1α-FLPXrc-[Chronos-GFP], AAV9-EF1a-DIO-hM4D(Gi)-mCherry, and AAV5-CamKIIα-hM4D(Gi)-mCherry were produced using the AAV Helper-Free System (Agilent Technologies) and the detailed methods of AAV production and purification were previously described[57,58]. AAV1-EF1α-Flpo (55637-AAV1)[36], AAVrg-EF1α-Flpo (55637-AAVrg)[36], AAV8-EF1a-fDIO-GCaMP6s (105714-AAV8, unpublished, deposited from Dr. Rylan Larsen lab), AAV9-hSyn-GCaMP6s (100843-AAV9)[41], AAVrg-EF1α-Cre (55637-AAVrg)[36], and AAV9-EF1α-FLEX-eNpHR3.0-EYFP (26966-AAV9)[37] were purchased from Addgene. AAV9-EF1α-fDIO-YFP, AAV9-EF1α-fDIO-PSAM⁴-GlyR-2A-mCherry[42,43], and AAV9-EF1α-fDIO-PSAM⁴-GlyR-2A-mCherry[39,43] were provided by the Dr. Makoto Tsuda laboratory.

The used viral titers were as follows: AAV2-retro-EF1α-Flpo, AAV2-retro-EF1α-Cre, AAV9-EF1a-eNpHR3.0-EYFP, and AAV5-CamKIIα-hM4D(Gi)-mCherry, $3 \times 10^{12}$ genome copies (GC)/ml; AAV9-EF1α-fDIO-YFP, AAV9-EF1α-fDIO-PSAM⁴-Gly-2A-mCherry, AAV9-EF1α-fDIO-hM3Dq-2A-mCherry, AAV8-EF1a-fDIO-GCaMP6s, and AAV_DJ-EF1α-FLPXrc-[Chronos-GFP], $5.0 \times 10^{12}$ GC/ml; AAV1-EF1α-Flpo, $1.5 \times 10^{13}$ GC/ml; AAV9-hSyn-Chronos-GFP, and AAV-hSyn-GCaMP6s, $1.5 \times 10^{12}$ GC/ml.

## Microinjections
We used the previously reported method with some modification[59]. Mice were deeply anesthetized with medetomidine hydrochloride (0.3 mg/kg, Domitol, Meiji Seika Pharma), midazolam (4 mg/kg, Dormicum, Astellas

Pharma) and butorphanol (5 mg/kg, Vetorphale, Meiji Seika Pharma), and the head of the mice was fixed in a stereotaxic apparatus (SR-5M-HT, Narishige). rAAV solutions were unilaterally injected (approximately 250 nl in one site) into the ACC [rostrocaudal (RC): +1.0 mm, mediolateral (ML): 0.3 mm, dorsoventral (DV): 0.8 mm], CLA [RC: +1.0 mm, ML: 2.8 mm, DV: 3.5 mm], IC [RC: +1.0 mm, ML: 3.3 mm, DV: 3.5 mm], and ventrolateral Striatum [RC: +1.0 mm, ML: 2.3 mm, DV: 3.5 mm]. Retrobeads (10% dilution, 300 nl, 1RX, LUMAFLUOR) were unilaterally injected into the ACC or CLA. We used virus-injected mice 3 weeks or more after the last injection of AAV vectors and Retrobeads-injected mice 1–2 weeks after the injection for further analysis.

## Immunohistochemistry
Immunohistochemical experiments were performed according to the methods described in our previous study[59]. Mice were deeply anesthetized by i.p. injection of medetomidine hydrochloride (0.3 mg/kg), midazolam (4 mg/kg), and butorphanol (5 mg/kg) and perfused transcardially with phosphate-buffered saline (PBS), followed by ice-cold 4% paraformaldehyde (PFA)/PBS. For examination of c-Fos immunoreactivity, we applied mechanical stimulation to the right paw repetitively with 2 g von Frey filament (North Coast Medical) for 15 min to induce licking or biting behavior, then perfused the animal 120 min after the mechanical stimulation as described above. The brains were removed, postfixed in the same fixative overnight at 4 °C, and placed in 30% sucrose solution for two overnight at 4 °C. Transverse brain sections (50 μm thick) were made and immunostained. The primary and secondary antibodies used are listed below.

Primary antibodies: polyclonal rabbit anti-c-Fos (sc-52, 1:500, Santa Cruz), polyclonal chicken anti-NeuN (266 006, 1:1000, Synaptic Systems), and polyclonal rabbit anti-RFP (PM005, 1:500, MBL Life Sciences)

Secondary antibodies: donkey anti-rabbit Alexa Fluor 647 (AB_2492288, 1:500, Jackson ImmunoResearch) and donkey anti-chicken Alexa Fluor 488 (AB_2340375, 1:1000, Jackson ImmunoResearch).

Immunofluorescent images were obtained with a confocal laser microscope (LSM780, Carl Zeiss). For quantification of fluorescent intensity of Chronos-GFP terminals, Retrobead positive cells, or c-Fos positive cells, 3–4 images containing the regions interested were acquired using a 10x objective and analyzed using Fiji (http://fiji.sc).

## In vitro slice whole-cell patch-clamp recording
We used the previous methods described in our previous study with some modifications[60,61]. Mice were deeply anesthetized with medetomidine hydrochloride (0.3 mg/kg), midazolam (4 mg/kg) and butorphanol (5 mg/kg), and the brain was quickly removed and placed into a cold high sucrose artificial cerebrospinal fluid (sucrose aCSF) (250 mM sucrose, 2.5 mM KCl, 2 mM CaCl₂, 2 mM MgCl₂, 1.2 mM NaH₂PO₄, 25 mM NaHCO₃ and 11 mM glucose). Coronal brain slices (300 μm thick) were cut using a vibrating microtome (NLS-MT, Dosaka), and then the slices kept in oxygenated artificial cerebrospinal fluid (aCSF) solution (125 mM NaCl, 2.5 mM KCl, 2 mM CaCl₂, 1 mM MgCl₂, 1.25 mM NaH₂PO₄, 26 mM NaHCO₃ and 20 mM glucose) at room temperature (22–25 °C) for at least 30 min.

Individual slices were then put into a recording chamber which was continuously perfused with ACSF solution at room temperature. Patch-clamp recordings were made from single neurons visualized with infrared differential interference contrast optics (BX50WI, Olympus). The patch pipettes (4–7 MΩ) were filled with an internal solution (K-gluconate 135, CaCl₂ 0.5, MgCl₂ 2, KCl 5, EGTA 5, 5 Mg-ATP, and HEPES, 0.2% Neurobiotin, pH 7.2 adjusted with KOH). Signals were amplified with Multi-Clamp 700 A amplifier and pCLAMP 10.4 acquisition software (Molecular Devices, USA) and digitized with an analog-to-digital converter (Digidata 1321 A, Molecular Devices), stored on a computer using a data acquisition program (Clampex version 8.2, Molecular Devices) and analyzed using a software package (Clampfit version10.4, Molecular Devices).

Light stimulation (5 mW/mm$^2$, 10 ms for CLA recording, 20 ms for ACC recording) was applied every 10 s through an optical fiber with a 200 μm tip width placed near the recording site. Light-evoked EPSCs were recorded in the voltage–clamp mode at −70 mV. The membrane potentials were recorded in the current-clamp mode, and the discharge pattern of the recorded neurons was examined by passing hyperpolarizing and depolarizing current pulses through the recording electrode from the resting membrane potential.

Following the completion of whole-cell patch–clamp recording, the slices were fixed overnight in 4% PFA/PBS at 4 °C to reveal the morphological feature of the recorded cells. The slices were rinsed in PBS and were immersed in 30% ethanol for 30 min and then incubated overnight in Streptavidin CF405M (1:500, Biotium) or Streptavidin Alexa555 conjugate (1:500, S21381, Molecular Probes).

Drugs used were TTX (1 μM, FUJIFILM Wako Chemicals), 4-AP (200 μM, Tokyo Chemical Industry), CNQX (10 μM, Tocris Bioscience), varenicline tartrate (100 nM, Tocris Bioscience) and CNO (10 μM, Tocris Bioscience).

## Behavioral analysis

We used the previous methods described in our previous study with some modifications[59]. To assess mechanical hypersensitivity, mice were placed individually in an opaque acrylic box (6 × 6 × 6 cm) on a wire mesh and habituated for ~1 h to allow acclimatization to the experimental environment. Calibrated von Frey filaments (0.02–2.0 g, North Coast Medical) were then applied to the plantar surfaces of the hindpaws of mice from below the mesh floor, and the 50% paw withdrawal threshold was determined using the up–down method[52]. The von Frey test was carried out before and at 30 or 60 min after injection of CNO (3 mg/kg, Tocris) or varenicline (0.5 mg/kg, Tocris), respectively. In the optogenetic study, light stimulation (5–10 mW, 10 Hz, 10 s) was delivered to the mice attached with optical fiber (Φ1.25 mm black ceramic ferrule, 0.39 NA, L = 3 mm, R-FOC-BL400C-39NA, RWD) through a patch cable, and we applied von Frey stimuli to the mice hindpaws without or during light stimulation.

To analyze the contralateral pain modulation during attending behaviors to the unilateral side, we applied a repetitive 2 g von Frey stimulation to the unilateral hindpaw until the mice exhibited attending behaviors for more than 3 s. The von Frey repetitive stimulation (33.0 ± 4.7 times, ~2 Hz, n = 5) induced "lifting and guarding" (16.5 ± 5.1 s) or "licking and biting" (20.9 ± 4.5 s) to the stimulated paw. After the repetitive stimulation, we applied 1 or 2 g von Frey stimulation to the contralateral paw during either "lifting and guarding", "licking and biting" or the period when no such behaviors were observed (not attending) to the repetitively stimulated paw (in total 5 times for each). To examine the effects of chemogenetic inhibition of ACC$^{→contralateral CLA}$ neurons, we did the above tests 1 h after varenicline (0.5 mg/kg) or saline injection and 30 min after CNO (3 mg/kg) or saline injection. Formalin (5%, 20 μl, FUJIFILM) was intraplantarly injected, and 20-30 min after the injection, von Frey tests to the contralateral hindpaw were examined. Varenicline or saline was injected 30 min before formalin injection.

For the capsaicin test and capsaicin-induced mechanical hypersensitivity test, 10 μl of capsaicin (1.6 μg per paw, Nakalai Tesque, dissolved in 10% ethanol and 10% Tween-80; Sigma in PBS) was intraplantarly injected using 30-gage needle 20 min after varenicline (0.5 mg/kg) or saline injection, and the amount of time that the mice spent licking and biting to the injected paw was measured for 5 min. After 30, 60, 90, 120, or 180 min after the capsaicin injection, von Frey filaments were applied to the hindpaw nearby the capsaicin injection site[59] and to the contralateral paw.

## In vivo awake extracellular recording

Mice were deeply anesthetized with medetomidine hydrochloride (0.3 mg/kg), midazolam (4 mg/kg), and butorphanol (5 mg/kg), and the head of the mice was fixed in a stereotaxic apparatus (SR-5M-HT, Narishige). For unit recording and optical stimulation, small cranial holes (Φ of ~1 mm) were made over the ACC (AP: 0.5–1.5, ML: 0–1.0) and the CLA (AP: 0.5–1.5, ML: 2.5–3.0) and covered until the day of recording. The head plate (Narishige, CP-2) was affixed to the skull with dental cement (UNIFAST II, GC). After surgery, mice were singly housed and fixed to the experimental apparatus (MAG-1 and SR-5M-HT) and habituated to the head-fixed condition for ~1 h per day at least 5 times before recording.

On the day of recording, under anesthesia with isoflurane (1.5–2%), EMG recording wires were sutured to the upper forelimbs, the mouse head was fixed to the stereotaxic apparatus, and a reference wire electrode was placed near the cranial hole over the ACC. For photostimulation of the CLA, an optical fiber (Φ of 200 μm, Thorlabs) was inserted just above the right claustrum (AP: 1.0, ML: 2.8, DV: 3.2). After recovery of anesthesia for ~20 min, a tungsten recording microelectrode (10 MΩ, FHC) was inserted into the left (for antidromic recording) or right (for contralateral recoding) ACC (AP: 0.6–1.2, ML: 0.4, DV: 0.4–1.0) with a micromanipulator (SMM-100, Narishige) and allowed to settle for ~10 min before recording. After obtaining a recording, the electrode was lowered ~20–50 μm until we succeeded in recording from the next neurons.

Multi-unit activity and LFP were amplified and filtered (multi-units: 300 Hz–3 kHz, LFP: 0.1–50 Hz) with a differential extracellular amplifier (EX1, Dagan). EMG was amplified and filtered (100–300 Hz) with an EMG amplifier (EBA-100, UNIQUE MEDICAL). These signals were digitized with an analog-to-digital converter (Digidata 1440, Molecular Devices) and stored on a personal computer at 20 kHz (for EMG and unit recording) or 100 kHz (for MUA, EEG, and EMG) using pCLAMP 8 software (Molecular Devices).

Spikes were sorted based on principal components analysis with offline sorter software (Plexon), and each spike width was calculated by the trough-to-peak interval of the mean spike waveform. We defined units with spike widths of >0.45 ms and <0.35 ms as putative pyramidal neurons and interneurons, respectively[62]. EMG and LFP signals were analyzed with LabChart8 software (AD Instruments).

ACC$^{→contraCLA}$ neurons were identified if they exhibited moderate firing rate (less than 20 Hz) and elicited antidromic APs following a train of light stimulation (473 nm, 0.8–1.5 mW/mm$^2$, 3 ms, 10 pulses, 20 Hz) with constant latency (9.41 ± 0.36 ms, n = 33) and short jitter (0.183 ± 0.010 ms, n = 33)[27,28,63]. Mechanical stimuli were applied to the ipsi- and contralateral whisker pad for ~0.5 s with von Frey filaments (0.16 g or 0.6 g). Whisker stroking was applied using closed forceps (~0.5 s duration). The time stamps for indication of the stimulus timing were acquired using a custom-made manual TTL pulse generator. The averaged firing rates of mechanical responses were obtained for 2 s after the stimulation and compared to that for 2 s before stimulation. For the capsaicin application, 10 μl of capsaicin (1.6 μg, dissolved in 10% ethanol and 10% Tween-80, Sigma in PBS) or vehicle was intradermally injected to the cheek of mice using a 30-gage needle. To mark the electrode track, we stained the electrode with DiI solution (Vybrant™ DiI Cell-Labeling Solution, V22885, Thermo Fisher) before recordings in a part of the experiments.

To examine the effects of activation of ACC$^{→contraCLA}$ neurons on the contralateral ACC neuronal activities, the unit activity and LFP were recorded from the contralateral ACC, and blue light pulses (3–5 mW/mm$^2$, 20 ms, 4–40 times, 2–20 Hz, 2 s (for LFP analysis), or 100 times, 10 Hz, 10 s (for sensory response modulation analysis)) were delivered through the optical fiber placed above the contralateral CLA to stimulate axon terminals of ACC$^{→contraCLA}$ neurons. In the experiments for optical inhibition of CLA$^{→iptiACC}$ neurons, a continuous red light pulse (5 mW, 5 s) was delivered from 2 s before the start of the blue light stimulation (4 to 40 times, 2–20 Hz).

## Fiber photometry experiment

Mice were deeply anesthetized with medetomidine hydrochloride (0.3 mg/kg), midazolam (4 mg/kg), and butorphanol (5 mg/kg), and the head of the mice was fixed in a stereotaxic apparatus (SR-5M-HT,

Narishige). Small cranial holes (Φ of ~1 mm) were made over the ACC (AP: 0.5–1.5, ML: 0–1.0), and mice were implanted with a 400 μm core optical fiber (Φ1.25 mm black ceramic ferrule, 0.39 NA, $L = 1$ mm, R-FOC-BL400C-39NA, RWD) in the ACC. Fibers and head plates (Narishige, CP-2) were fixed with dental cement (UNIFAST II, GC). Animals were allowed to recover for ~7 days.

Fluorescent recordings were made using a Doric Lenses photometry system (iFMC4-G2_IE(400-410)_E1(460-490)_F1(500-550)_S). GCaMP6s were excited by a 465 nm light-emitting diode (LED, CLED_465, Doric Lenses) light continuously driven by an LED driver (LEDD_2, Doric Lenses), and the emission fluorescence signals passed from the brain were detected by the system, and recorded and digitized with an analog-to-digital converter (Digidata 1550B, Molecular Devices). The time stamps for indication of the stimulus timing were acquired using a custom-made manual TTL pulse generator. The fluorescent responses during repetitive stimulation to each hindpaw were obtained by averaging fluorescent intensity for 20 s during the stimulation and compared to that for 5 s before the stimulation.

## Statistics and reproducibility

Statistical analyses were performed using Prism 9 (GraphPad). All data are shown as the mean ± standard error of the mean (SEM). In the figure legends, we provide details on the sample numbers, statistical tests used, and the results of all statistical analyses for each experiment and all statistical comparisons. Statistical significance of differences was determined using one-way ANOVA with Tukey's multiple comparisons test, one-way ANOVA with Dunnett's multiple comparisons test, one-way ANOVA with Bonferroni's multiple comparisons test, one-way repeated measures ANOVA with Dunnett's multiple comparisons test, one-way repeated measures ANOVA with Tukey's multiple comparisons test, two-way repeated measures ANOVA with Bonferroni's multiple comparisons test, two-tailed paired *t*-test, or two-tailed unpaired *t*-test. Differences were considered significant at $p < 0.05$. Statistical details for the experiments (including exact n values, statistical tests used, *p* and *F* values) are summarized in the figure legends or Supplementary Data.

## Reporting summary

Further information on research design is available in the Nature Portfolio Reporting Summary linked to this article.

## Data availability

All the original data that support the findings of this study are available from the corresponding author upon request. Numerical source data underlying graphs in the paper can be found in the Supplementary Data file.

## Code availability

No code was used for the study, and software information was provided in the Method section.

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

## Acknowledgements

We thank the joint-use research facilities and the Center for Comparative Medicine of Hyogo Medical University for the use of facilities and appreciate the technical support from Dr. Yasuhiro Tanaka (Tamagawa University, Japan). This work was supported by the following grants. JSPS KAKENHI Grant JP20H04043 and JP23K18441 (H.F.), JSPS KAKENHI Grant JP20K16133 and JP22K15206 (K. Koga), Hyogo Innovative Challenge grant (H.F.), Hyogo College of Medicine Grant for Research Promotion 2021 and 2023 (K. Koga), AMED under Grant Number JP23gm1510013 (H.F.), MEXT Promotion of Distinctive Joint Research Center Program Grant Number JPMXP0723833162 (K. Koga), JSPS KAKENHI Grant Number JP 22H04922 (AdAMS), Takeda Science Foundation (K. Koga), the 45th Nakatomi Science Foundation (K. Koga), the Uehara Memorial Foundation (K. Koga), and the Naito Foundation (K. Koga).

## Author contributions

K. Koga conceived this project, designed experiments, performed almost all experiments, analyzed the data, and wrote the paper. K. Kobayashi and M.T. provided critical materials and advised on experimental procedures. A.E.P. provided critical advice on data interpretation and experimental design and wrote the paper. H.F. conceived this project, supervised the overall project, designed experiments, and wrote the paper.

## Competing interests

The authors declare no competing interests.
