## [Peer review file · Communications Biology]

Reviewers' comments:

Reviewer #1 (Remarks to the Author):

This manuscript suggests that a circuit from the ACC to the contralateral claustrum can inhibit activity in the contralateral ACC. A series of experiments were carried out to suggest that this pathway might have a role in sensory responses and pain behavior. The authors perform some very nice in vivo electrophysiology. Unfortunately, the role of the ACC to contra claustrum pathway was not actually investigated, as nowhere in the manuscript were claustrum neurons manipulated directly.

Below are the are the main issues with this manuscript.

Figure 2 shows the activity of ACC – contra claustrum cells. This figure shows very well that ACC neurons are activated by contralateral stimulation of the body, which is not surprising given the decades of work on the organization of somatosensory coding and lateralization. This figure does not address anything directly about the claustrum, only that these cells (probably) project to the claustrum. But what these action potentials do when they arrive in the claustrum is not solved by the manuscript. The interpretation of the data, suggesting that this circuit in the ACC suppresses the contralateral ACC through claustrum is confounded by antidromic activation of the ACC itself. Activating deep layers of the cortex also generates a similar form of cortical silencing (see Bortone et al 2014 Neuron) as activating the claustrum, so one cannot rule this out without additional experiments.

Figure 3 shows slice recordings mainly confirming the work from Solange Brown's lab, and the Silberberg lab showing activation of both interneurons and excitatory cells in the contralateral claustrum.

Figure 4 shows that activating ACC projections to the contralateral claustrum evokes inhibition in the contralateral ACC. The authors claim this is due solely to the activation of the claustrum. However, this could be due to antidromic activation of the ipsilateral ACC, as they show that these neurons are robustly activated by this same stimulation (in their Figure 2). Therefore, this is a massive caveat for their interpretation.

Figure 4K-Q show that chemogenetic activation of ACC cells projecting somewhere around the contralateral claustrum change paw withdrawal behavior. Again, there is nothing that can be said about claustrum neural activity in these figure panels, because claustrum cells were not manipulated. These figure panels can speak to what the ACC does, but not the claustrum.

The same caveats remain for all of Figure 5 and 6. Nowhere in these figures do we learn about what the claustrum does. However, there is great data regarding the ACC. Only ACC cells were manipulated, and therefore the only thing that can be claimed in this manuscript is that ACC cells play a role in these sensory and pain behaviors.

Another overarching caveat is the use of retro AAVs into the claustrum. Given the close proximity of the striatum and insula to the claustrum it would be important to show that only the claustrum received the retro aav injection.

Reviewer #2 (Remarks to the Author):

The anterior cingulate cortex (ACC) is a crucial hub for coordinating executive functions. During pain, it is known that the ACC responds to noxious sensory input and integrates those input to guide appropriate behaviours. The claustrum is strongly reciprocally connected to the ACC and is proposed to be involved in sensory processing and attentional allocation. Because the ACC is involved in modulating pain responses and receives input from the claustrum, Koga et al. examined the functions of an ACC-Cla-ACC loop during pain perception. The authors found that:

- 1) ACC neurons receive mechanosensory input from the contralateral side and transmit that information to the contralateral claustrum (ipsilateral to the pain stimulus).
- 2) The activation of this contralaterally-projecting ACC-Cla circuit, inhibited the ACC on the opposite side via the recruitment of FS-IN.
- 3) The suppression of the (ipsilateral) ACC reduced the likelihood to induce nociceptive behaviour on the “wrong side” of the body, helping to increase the chances to induce appropriate behaviours to the incoming noxious signals in a body-side-specific manner.
- 4) Finally, they showed that interference with lateralized cortico-claustrum-cortical circuits results in pain hypersensitivity on the side of the body that is not receiving the noxious stimulus input, supporting the hypothesis that claustrum-cortical circuits are involved in the modulation of ACC-mediated executive functions to guide appropriate behaviours.

The major scientific contribution of this study is the finding that claustral activity is modulating pain sensation (Figures 5 and 6). Most interestingly are the findings in Figure 6, indicating that claustral activity normal is involved in suppressing pain sensation in one hemisphere of the body, indicating that the claustrum could be a possible future target to suppress/modulate pain sensation. In general, the experiments were well performed and employed several anterograde or retrograde tracing methods to label the relevant cortico-claustrum-cortical circuits.

Overall, the conclusions stated in the paper are sound. However, I have a concern about the claim (line 314) that “their findings define a novel mechanism by which the ACC influences the processing of spatially distinct (lateralized) and competing sensory input to prioritize and resolve behavioral conflicts”. In fact, many other studies have made this point; an incomplete listing of relevant papers includes the following:

- 1) The preferential inputs from contralateral M1 and ACC to the claustrum, and ipsilateral outputs from the claustrum to ACC was already shown by: Smith et al., 2014 (PMID: 24904315); Wang et al., 2017 (PMID: 24904315); Zingg et al., 2018 (PMID: 24904315)
- 2) Electrophysiological characterization of ACC-Cla connections: Chia et al., 2020 (PMID: PMID: 32531275); White et al., 2018 (PMID: 29298436)
- 3) Claustrum-cortical effects on ACC activity/cortical recruitment of IN: Jackson et al., 2018 (PMID: 30122374); Narikiyo et al., 2020 (PMID: 32393895), McBride et al., 2022 (PMID: 36368317)
- 4) Involvement of claustrum in distractor suppression (in your case< sensory inputs from the opposite body side): Atlan et al., 2018 (PMID: 30122531)
- 5) Claustrum activity decodes lateralized information about upcoming movements: Chevee et al., 2022 (PMID: 34863367)

While many aspects of the manuscript have already been established by other studies, the current manuscript makes a genuinely novel contribution by implicating the claustrum in pain processing. The authors were able to comprehensively reproduce these findings in a single study. The strength of

this study is the functional confirmation of specific, hemisphere-crossing cortico-claustrum-cortical loops, and the involvement of those loops to optimize ACC signal processing and guide a situation-appropriate behavioural response to pain. The most striking finding is the involvement of the claustrum in reducing pain hypersensitization on the contralateral side, implicating that the claustrum could be a potential target for future pain modulation studies. For this reason, I support the publication of this manuscript.

Beyond the question of proprietary claims of novelty (above), the authors should address the following issues:

- Line 56: "ACC neurons preferentially project to the contralateral claustrum"; Is your generalization about all ACC connections, or are you only emphasizing the claustral connections? If the latter is the case, please make that clear. For example: "It is known that the ACC connects more strongly to the contralateral than the ipsilateral claustrum (add reference)".
- Line 102: 1 latency of 10 ms seems a bit long for antidromic activation; if axonal conduction velocity is 10 m/s, then an action potential can propagate 1 cm in 1 ms. This is much greater than the length of ACC neuron axons, so it is not clear whether the activity evoked by photostimulation is genuine antidromic firing, or results from a synaptic connection.
- Line 111: Title is misleading. At this point it is not clear what is cause or effect: Are ACC neurons more active because of the attending wiping behaviour, or is the behaviour increased due to the increased activity of ACC-Cla neurons that drives wiping?
- Line 134: CLA neurons projecting to the ACC do not uniformly exhibit bursting activity; see Fig. 3 of Chia et al. (2017) <http://dx.doi.org/10.1080/20023294.2017.1357412>.
- Line 144: Are FS-IN the only targeted neuron type? How about layer 1 interneurons...they should be easily patchable and you could gain some info about other IN types? Have you done any recordings from the upper layers and do you see layer differences in general?
- Line 162 – the authors should explain that they are measuring membrane capacitance to estimate cell surface area.
- Lines 180ff: An excitatory input to the ACC drives both interneurons and projection neurons, however, PPR differences indicate a faster decay for INs when compared to more stable Cla->PN connections. Wouldn't the inhibition only last for a short while and then the excitatory effects would dominate, resulting in a power increase in the range of your stimulation frequency (basically your power is increased due to Chronos activation of claustral axons [stimulus artifact] rather than a truly intrinsic generation of slow-wave oscillations??). Are you able to observe a selective increase in slow-wave power at higher stimulus frequencies?
- Line 195 – some explanation should be given for the shift from optogenetic to chemogenetic activation strategies. Both approaches are good, but it is odd to see an unexplained switch from one to the other.
- Line 235: Why shift c-Fos measurements for Cla into the supplementary data, after you show cFos measurements for the ACC just one sentence before?
- Line 328: attending behaviour versus simple motor effects? How to differentiate attending versus motor behaviour: Did ACC-contra Cla neurons show generally an increased activity when the mouse is moving, or was the increase specific for pain-attending movement?
- Line 337f: Mechanisms of claustral activity not identified? As mentioned before, there is ample evidence in the literature that provided data for the existence of an ACC-Cla circuit (ACC-Cla: White et al., 2018, Chia et al., 2020) or the specific recruitment of the claustrum during higher cognition (Atlan, 2018, Terem, 2020)
- Line 350: probably should add a statement that this is in line with previous findings

- Line 351: "attention or affect"? What do you mean?
- Line 410: 250 nl injection volume is relatively large, especially for claustral injections - you might have also labelled insular cortex neurons. 1) While insular cortex-ACC projections are rare, they do exist. 2) As the insula is involved in pain processing, optogenetic activation of direct ACC-ins Ctx circuits might contribute to pain processing. Have you checked for any Chronos-labelled insular cortex neurons projecting to the ACC (retroflip ACC- Chronos in claustrum)? Please provide some high-resolution images of the upper insular cortex layers next to the labelled claustrum

I also have some editorial comments for the figures:

Figure 1

- a. Minor: Panel B: Do you have any examples of a less leaky viral injections?
- b. Minor: Panel E-G: due to the spread of retrobeads into the INS cortex, and clearly the endopiriform cortex, the performed cell counts reflect connections to the surrounding areas of the claustrum. The quantification is unlikely to reflect network differences that are limited to the claustrum and hence, should be removed. Also, tiling effect in panel E is disturbing.
- c. Lastly, panel C clearly indicates that ACC projects to both hemispheres, indicating claustral modulation on both sides, indicating that your work only depicts a part of the network. This should be at least be mentioned in the discussion.

Figure 2

- a. Panel A: As the contralateral motor cortex also projects to the claustrum, please provide a figure showing that the recording electrode was in the ACC not the motor cortex... otherwise motor signals could have been interpreted as ACC-> Cla signals
- b. Panel B: Your optical fiber was close to the strongly labelled external capsule: how did you prevent stimulating passing axons that could be interpreted as claustrum-targeting in ACC electrode recordings (false-positives)?
- c. Panel D-F: are those all the recorded units? What's going on top of the plot? There are neurons that seem to fire at a much higher frequency. These neurons don't seem to be affected. Are they INs (spike width) or PN? If these are INs, they shouldn't be included and counted as Cla-projecting (unless you can confirm identity by IHC/marker expression and retrobead/marker expression.
- d. Panel H: What is shown on the top trace (label missing).. how are unit activities indicated in h? Sometimes a large spike is indicated, sometimes not...Please indicate the detection threshold in the top trace to clarify when unit activity was detected and plotted in row 2 ("unit"). What's that rhythmic signal in EMG? Please add short comment non-EMG people into the legend.
- e. Panel H/I: Unit responses reflect movement on the contralateral body side. Do you also have unit recordings from the ipsilateral side? Is the ACC silent for ipsilateral movements? Basically, how specific does the ACC encode lateralized movements?
- f. As shown in panel D-F, the responses of contralateral-projecting ACC->Cla units reflects reliably contralateral stimulus input (that reaches a ceiling effect quickly), while ipsilateral mechanical stimulations depend on the strength of the stimulus (responses ACC-> Cla units) 0.2g< 0.6g< whisker stroking). For H-J. What was the used strength of the mechanical whisker stimulation? For ipsilateral stimuli (j), was the chosen stimuli to weak (0.16g) to elicit a response in the first place (see ipsilateral stimuli at 0.16g in panel d).

Figure 3:

- a. Panel D: Why are there multiple EPSCs? Explanation for multiple EPSCs is missing
- b. Panel H: Looks like some ipsilateral axons also target contra and ipsilateral striatum and ventral brain areas ...are those blood vessel artifacts or actually axons?

c. Panels J-P: You are showing two representative neurons, not the morphological features. A graphical representation of connectivity percentage (EPSCs) and percentage of AP triggered would be great. Also, because PNs might have higher Rms than FS INs, a lower EPSC might result in comparable or higher depolarizations than observed in FS INs. What is the average EPSP amplitude for INs vs PNs? And lastly, as you observed a nearly significant difference of the paired pulse ratio, how do these depolarization differences translate for a train of stimuli (10Hz, 20Hz or even higher)? It seems like the second EPSC for PNs is higher than for INs... Could that imply a change in the preferred population of recruited neurons (initial recruitment INs, later PNs)? Any additional info's could potentially help to settle the ongoing discussion of claustral effects being either excitatory or inhibitory.

Figure 4:

- a. Panel B: color scale is missing; also...after each stimulation you can observe an increase in the power range between 10 and 20Hz...is that significant?
- b. Panel E and figure legend: The effect on PNs seem to be bidirectional: first an inhibition, followed by a period of increased activity around 200ms after the stimulus.
- c. Panel G-I and J: compare the peak frequencies. In general, the frequency peak in the average PTSH in the bottom of G-I is higher than the values in J. Why? Also, how was J quantified? Seems like peak amplitude wise, panel G responses showed no differences in the PN firing rate but are indicated as significantly lower in J? The panels E and G-J don't support each other and the observed inhibitory responses in G-I only start to be effective after an initial increase of AP firing. How do you explain those effects (fast unit inhibition in E, population activation in G-I even in the presence of CLA \rightarrow ACC activation; late inhibitory effects)
- d. Panel N: The chemogenetic activation of neurons appear to be constant. That is troublesome as you lose the ability to decode time-sensitive information that are probably critical under real-life conditions (acute pain events).

Figure 5:

- a. Panel 5d: statistical test results for hind paw results were shown in main text, the attending behaviour was not shown (is just one more line)... why?
- b. Panel 5j: AAV9 labels anterograde and retrograde cell populations, chemogenetic effects are not region-specific. Do you have brain slide images for mCherry expression in the claustrum to confirm that you are not inhibiting the claustrum directly.
- c. Compare 5f and 5j: both panels show retrograde labelling of CLA projecting ACC neurons. Why is it that in panel 5f, the labelled neurons are more superficial, while in 5j most claustral projecting neurons are in deep layers?

Figure 6:

- a) In general, after establishing the ACC \rightarrow contralateral CLA \rightarrow ipsi ACC \rightarrow ipsi ACC loop, it would be good to graphically represent this loop including the postulated effects of mechanical sensation and ACC activity.

re: comments of Reviewer 1

This manuscript suggests that a circuit from the ACC to the contralateral claustrum can inhibit activity in the contralateral ACC. A series of experiments were carried out to suggest that this pathway might have a role in sensory responses and pain behavior. The authors perform some very nice in vivo electrophysiology. Unfortunately, the role of the ACC to contra claustrum pathway was not actually investigated, as nowhere in the manuscript were claustrum neurons manipulated directly.

Reply: We thank the reviewer 1 for the detailed suggestions and comments. We have conducted additional experiments and analysis to confirm the involvement of claustrum (CLA) in our original observations, and included these results and description that address the points raised by him/her in the revised manuscript.

Below are the are the main issues with this manuscript.

Figure 2 shows the activity of ACC – contra claustrum cells. This figure shows very well that ACC neurons are activated by contralateral stimulation of the body, which is not surprising given the decades of work on the organization of somatosensory coding and lateralization. This figure does not address anything directly about the claustrum, only that these cells (probably) project to the claustrum. But what these action potentials do when they arrive in the claustrum is not solved by the manuscript. The interpretation of the data, suggesting that this circuit in the ACC suppresses the contralateral ACC through claustrum is confounded by antidromic activation of the ACC itself. Activating deep layers of the cortex also generates a similar form of cortical silencing (see Bortone et al 2014 Neuron) as activating the claustrum, so one cannot rule this out without additional experiments.

Reply: As pointed by the reviewer, our original manuscript did not directly manipulate the CLA activity. Therefore, we performed additional experiments to manipulate claustral activity using optogenetic and chemogenetic methods, relating to the reviewer's following comments as well. Especially, we performed optogenetic inhibition experiments, where we expressed inhibitory opsin, halorhodopsin (eNpHR3.0), in ACC-projecting claustral (CLA^{→ipsiACC}) neurons to inhibit their activity using a retrograde strategy. We found that the EEG delta power increase induced by optogenetic activation of the terminals of ACC neurons projecting to the contralateral CLA (ACC^{→contraCLA} neurons) was suppressed by inhibition of the contralateral CLA^{→ipsiACC} neurons. This result suggests that activation of the ACC-projecting CLA neurons by inputs from the ACC^{→contraCLA} neurons is involved in the following SW induction. We have included these data in the revised Supplementary Fig. 5 and in the Result section of the revised manuscript (line 202–217).

Figure 3 shows slice recordings mainly confirming the work from Solange Brown's lab, and the Silberberg lab showing activation of both interneurons and excitatory cells in the contralateral claustrum.

Reply: As pointed by the reviewer, because the former part of the original figure 3 (the original Fig. 3a-k) have been reported in previous works, we moved them to the revised Supplementary Fig. 2. However, how the CLA neurons receiving contralateral ACC inputs ($CLA \leftarrow \text{contraACC}$ neurons) project to the ipsilateral ACC have not been investigated. Therefore, we decided to keep the later part as the main Figure (the revised Fig. 3), where we also added the recording data from superficial layers relating to the following comments from the reviewer 2 (Fig. 3 k–v, line 170–180).

Figure 4 shows that activating ACC projections to the contralateral claustrum evokes inhibition in the contralateral ACC. The authors claim this is due solely to the activation of the claustrum. However, this could be due to antidromic activation of the ipsilateral ACC, as they show that these neurons are robustly activated by this same stimulation (in their Figure 2). Therefore, this is a massive caveat for their interpretation.

Reply: As we have mentioned in the former reply, we performed additional experiments to directly manipulate claustral activity using optogenetic methods to confirm the participation of the contralateral CLA in the effects induced by activation of $ACC \rightarrow \text{contraCLA}$ neurons. Specifically, we performed optogenetic inhibition experiments using halorhodopsin to inhibit activity of $CLA \rightarrow \text{ipsiACC}$ and found that inhibition of $CLA \rightarrow \text{ipsiACC}$ neurons significantly reduced the EEG delta power increase in the contralateral ACC induced by optogenetic activation of $ACC \rightarrow \text{contraCLA}$ neurons. This result suggests that activation of the ACC-projecting CLA neurons by inputs from the $ACC \rightarrow \text{contraCLA}$ neurons is involved in the following SW induction. We have included these data in the revised Supplementary Fig. 5 and in the Result section of the revised manuscript (line 202–217).

Figure 4K-Q show that chemogenetic activation of ACC cells projecting somewhere around the contralateral claustrum change paw withdrawal behavior. Again, there is nothing that can be said about claustrum neural activity in these figure panels, because claustrum cells were not manipulated. These figure panels can speak to what the ACC does, but not the claustrum.

Reply: Relating to the comments from the reviewer 2, we replaced the original chemogenetic data (the original Fig 4k–q) with the optogenetic data (the revised Fig. 4k–n), but we confirmed that activation of the $ACC \rightarrow \text{contraCLA}$ neurons inhibits the behavioral responses to ipsilateral mechanical stimulation. Furthermore, to confirm the involvement of the CLA in these observations, we performed an anterograde experiment to express Chronos-GFP in $CLA \leftarrow \text{contraACC}$ neurons in the right CLA, downstream target of the left $ACC \rightarrow \text{contraCLA}$ neurons. Optogenetic activation of the $CLA \leftarrow \text{contraACC}$ neurons mimicked the above behavioral effects induced by activation of $ACC \rightarrow \text{contraCLA}$ neurons. These results indicate that the contralateral CLA is involved in the behavioral changes induced by activation of $ACC \rightarrow \text{contraCLA}$ neurons. We have included these data in the revised main figure (Fig. 4k–q), and in the Result section of the revised manuscript (line 226–239), and moved the chemogenetic data to the

revised supplementary information (Supplementary Fig. 6).

The same caveats remain for all of Figure 5 and 6. Nowhere in these figures do we learn about what the claustrum does. However, there is great data regarding the ACC. Only ACC cells were manipulated, and therefore the only thing that can be claimed in this manuscript is that ACC cells play a role in these sensory and pain behaviors.

Reply: Regarding figure 5 and 6, we performed additional experiments to confirm participation of the contralateral CLA in the behavioral alterations induced by inhibition of ACC^{→contraCLA} neurons using an anterograde viral transduction method, where we introduced PSAM4, an inhibitory chemogenetic actuator, in the right CLA_{←contraACC} neurons. Inhibition of the CLA_{←contraACC} neurons reproduced the behavioral data that we showed in figure 5 and 6, confirming involvement of the contralateral CLA in these original observations. We have included these data in the revised Supplementary Fig. 8 and in the Result section of the revised manuscript (line 314–320, 350–354).

Another overarching caveat is the use of retro AAVs into the claustrum. Given the close proximity of the striatum and insula to the claustrum it would be important to show that only the claustrum received the retro aav injection.

Reply: In our tracing data (shown in the revised Fig. 1e–g, Supplementary Fig. 1e–h), only when we injected to the retrograde tracer to the CLA, we could obtain efficient retrograde labeling, while we did not observe such efficient labeling in the contralateral ACC when we injected it to adjacent areas, insular cortex (IC) or ventrolateral striatum (vlStr) (the revised Supplementary Fig. 1e–h). In parallel with these observations, the terminals of ACC^{→contraCLA} neurons or those of ACC pyramidal neurons were not observed the adjacent areas (the revised Fig. 5j, *left panel*, Supplementary Fig. 1a–d). Therefore, we concluded that our results of retrograde experiments were mainly mediated by manipulation of the ACC neurons projecting to the contralateral CLA, even though it is difficult to completely control the retro AAV spread. We have included the above data in the revised Supplementary Fig. 1, and in the Result section of the revised manuscript (line 93–99).

re: comments of Reviewer 2

The anterior cingulate cortex (ACC) is a crucial hub for coordinating executive functions. During pain, it is known that the ACC responds to noxious sensory input and integrates those input to guide appropriate behaviours. The claustrum is strongly reciprocally connected to the ACC and is proposed to be involved in sensory processing and attentional allocation. Because the ACC is involved in modulating pain responses and receives input from the claustrum, Koga et al. examined the functions of an ACC-Cla-ACC loop during pain perception. The authors found that:

- 1) ACC neurons receive mechanosensory input from the contralateral side and transmit that information to the contralateral claustrum (ipsilateral to the pain stimulus).*
- 2) The activation of this contralaterally-projecting ACC-Cla circuit, inhibited the ACC on the opposite side via the recruitment of FS-IN.*
- 3) The suppression of the (ipsilateral) ACC reduced the likelihood to induce nociceptive behaviour on the “wrong side” of the body, helping to increase the chances to induce appropriate behaviours to the incoming noxious signals in a body-side-specific manner.*
- 4) Finally, they showed that interference with lateralized cortico-claustrum-cortical circuits results in pain hypersensitivity on the side of the body that is not receiving the noxious stimulus input, supporting the hypothesis that claustrum-cortical circuits are involved in the modulation of ACC-mediated executive functions to guide appropriate behaviours.*

The major scientific contribution of this study is the finding that claustral activity is modulating pain sensation (Figures 5 and 6). Most interestingly are the findings in Figure 6, indicating that claustral activity normal is involved in suppressing pain sensation in one hemisphere of the body, indicating that the claustrum could be a possible future target to suppress/modulate pain sensation. In general, the experiments were well performed and employed several anterograde or retrograde tracing methods to label the relevant cortico-claustrum-cortical circuits.

Overall, the conclusions stated in the paper are sound. However, I have a concern about the claim (line 314) that “their findings define a novel mechanism by which the ACC influences the processing of spatially distinct (lateralized) and competing sensory input to prioritize and resolve behavioral conflicts”. In fact, many other studies have made this point; an incomplete listing of relevant papers includes the following:

- 1) The preferential inputs from contralateral M1 and ACC to the claustrum, and ipsilateral outputs from the claustrum to ACC was already shown by: Smith et al., 2014 (PMID: 24904315); Wang et al., 2017 (PMID: 24904315); Zingg et al., 2018 (PMID: 24904315)*
- 2) Electrophysiological characterization of ACC-Cla connections: Chia et al., 2020 (PMID: PMID: 32531275); White et al., 2018 (PMID: 29298436)*

3) *Claustro-cortical effects on ACC activity/cortical recruitment of IN: Jackson et al., 2018 (PMID: 30122374); Narikiyo et al., 2020 (PMID: 32393895), McBride et al., 2022 (PMID: 36368317)*

4) *Involvement of claustrum in distractor suppression (in your case < sensory inputs from the opposite body side): Atlan et al., 2018 (PMID: 30122531)*

5) *Clastrum activity decodes lateralized information about upcoming movements: Chevee et al., 2022 (PMID: 34863367)*

While many aspects of the manuscript have already been established by other studies, the current manuscript makes a genuinely novel contribution by implicating the claustrum in pain processing. The authors were able to comprehensively reproduce these findings in a single study. The strength of this study is the functional confirmation of specific, hemisphere-crossing cortico-claustrum loops, and the involvement of those loops to optimize ACC signal processing and guide a situation-appropriate behavioural response to pain. The most striking finding is the involvement of the claustrum in reducing pain hypersensitization on the contralateral side, implicating that the claustrum could be a potential target for future pain modulation studies. For this reason, I support the publication of this manuscript.

Reply: We are pleased that Reviewer 1 finds that our study “makes a genuinely novel contribution by implicating the claustrum in pain processing”. We thank him/her for the detailed suggestions and comments, and appreciate his/her support for the publication of our study. We have conducted additional experiments and analysis, and included these results and description that address the points raised by him/her in the revised manuscript.

Beyond the question of proprietary claims of novelty (above), the authors should address the following issues:

- Line 56: “ACC neurons preferentially project to the contralateral claustrum”; Is your generalization about all ACC connections, or are you only emphasizing the claustral connections? If the latter is the case, please make that clear. For example: “It is known that the ACC connects more strongly to the contralateral than the ipsilateral claustrum (add reference)”.

Reply: Following the reviewer’s suggestion, we have changed the description and added reference (line 56–57).

- Line 102: 1 latency of 10 ms seems a bit long for antidromic activation; if axonal conduction velocity is 10 m/s, then an action potential can propagate 1 cm in 1 ms. This is much greater than the length of ACC neuron axons, so it is not clear whether the activity evoked by photostimulation is genuine antidromic firing, or results from a synaptic connection.

Reply: As pointed by the reviewer, it is true that the latencies of antidromic activation seems to be a bit long, but the latencies of optogenetic antidromic activation were reported as 7.31 ± 0.32 ms or 4–

17 ms depending on projection distances in previous studies (Jennings et al., Nature, 496, 224–228, 2013, and Ciochi et al., Science, 348, 560-3, 2015), and our results (9.41 ± 0.36 ms) are equivalent to those values. Furthermore, the jitter of our study is also equivalent to those in the previous studies. It is unknown why the latencies of optogenetic antidromic activation are longer than the ones calculated from their velocity and distance, but even in the slice recording experiment, the latency of optogenetic antidromic activation was long, which reported as 6.07 ± 1.02 ms (Gruver et al., Front. Synaptic Neurosci, 11:31, 2019). Further studies are needed to explain why the latency of optogenetic antidromic activation becomes longer than that estimated from the axonal conduction velocity.

- Line 111: Title is misleading. At this point it is not clear what is cause or effect: Are ACC neurons more active because of the attending wiping behaviour, or is the behaviour increased due to the increased activity of ACC-CLA neurons that drives wiping?

Reply: Even though a lot of studies have suggested that ACC activity represents multiple aspects of pain and associated action planning (see discussion section in the revised manuscript (line 385–391), we couldn't conclude what is cause or effect from our results. Therefore, as the reviewer suggested, we changed the description, and just mentioned about the relationship between activities of ACC \rightarrow contraCLA neuron and the following behaviors. Further studies are needed to identify the underlying mechanisms of this facilitation. We have changed the title (line 118–119), and have added the descriptions in the discussion section of the revised manuscript (line 391–393).

- Line 134: CLA neurons projecting to the ACC do not uniformly exhibit bursting activity; see Fig. 3 of Chia et al. (2017) <http://dx.doi.org/10.1080/20023294.2017.1357412>.

Reply: We apologize for our misleading usage of “burst firing” because we did not differentiate a burst firing and strongly adapting firings in the original manuscript. Because all the neuron we recorded exhibit strongly adapting firings including a burst firing we have corrected “burst firing” to “strongly adapting firings”, and added the reference in the corresponding text of the revised manuscript (line 142–143).

- Line 144: Are FS-IN the only targeted neuron type? How about layer I interneurons...they should be easily patchable and you could gain some info about other IN types? Have you done any recordings from the upper layers and do you see layer differences in general?

Reply: As requested by the reviewer, we performed additional recording from layer I interneurons and layer II pyramidal neurons. They also received inputs from CLA \leftarrow contraACC neurons, but they did not exhibit action potential firings, suggesting that fast-spiking interneurons are the only cells to exhibit AP firings by light evoked inputs from CLA \leftarrow contraACC neurons. We have included these data in the revised main figures (Fig. 3k–v) and in the Result section of the revised manuscript (line 170–180).

- Line 162 – the authors should explain that they are measuring membrane capacitance to estimate cell surface area.

Reply: As requested by the reviewer, we have added the explanation for estimation of cell surface area by recording membrane capacitance (line 172–174).

- Lines 180ff: An excitatory input to the ACC drives both interneurons and projection neurons, however, PPR differences indicate a faster decay for INs when compared to more stable Cla->PN connections. Wouldn't the inhibition only last for a short while and then the excitatory effects would dominate, resulting in a power increase in the range of your stimulation frequency (basically your power is increased due to Chronos activation of claustral axons [stimulus artifact] rather than a truly intrinsic generation of slow-wave oscillations??). Are you able to observe a selective increase in slow-wave power at higher stimulus frequencies?

Reply: To investigate whether the activation of ACC^{→contraCLA} neurons at higher stimulus frequencies induce the EEG delta power increase as shown in the original data, we performed additional *in vivo* LFP recording experiments using light stimulations at 2, 5, 10, and 20 Hz. The EEG delta power was increased by all frequency stimulations. However, the EEG delta power increase at 20 Hz stimulation was significantly lower than those induced by the lower frequencies. It could be because the synaptic inputs to fast-spiking interneurons from the CLA decay faster, and the efficacy of the recruitment of their activation would be decreased in higher frequency. We have included these data in the revised Supplementary Fig. 4 and in the Result section of the revised manuscript (line 197–202)

- Line 195 – some explanation should be given for the shift from optogenetic to chemogenetic activation strategies. Both approaches are good, but it is odd to see an unexplained switch from one to the other.

Reply: Regarding the reviewer's following comment, we performed additional optogenetic experiments, and these results support our original hypothesis that the ACC^{→contraCLA} inhibit the ipsilateral sensory responses (contralateral to the fiber implantation). Furthermore, regarding a comment from the former reviewer, we also performed an anterograde experiment and confirmed that these behavioral changes were mediated by the contralateral CLA. We have replaced the chemogenetic data with the above optogenetic data in the revised main figure (Fig. 4k–q), and moved chemogenetic data to the revised Supplementary Fig. 6 and added descriptions of the optogenetic data in the Result section of the revised manuscript (line 226–239).

- Line 235: Why shift c-Fos measurements for Cla into the supplementary data, after you show cFos measurements for the ACC just one sentence before?

Reply: Due to the limitation of the space of the main figure, we decided to keep the result of c-Fos measurements for the CLA in supplementary information (the revised supplementary Fig. 7).

- Line 328: attending behaviour versus simple motor effects? How to differentiate attending versus motor behaviour: Did ACC-contra CLA neurons show generally an increased activity when the mouse is moving, or was the increase specific for pain-attending movement?

Reply: In the fiber photometry experiments and in vivo electrophysiological analysis, ACC^{→contraCLA} neurons did not exhibit any clear activities related to simple spontaneous movement, but they preferentially responded to the contralateral somatosensory inputs, and these responses were enhanced when they exhibit attending behaviors. Therefore, we have concluded that sensory responses in ACC^{→contraCLA} neurons are augmented when the mice exhibit attending behavior, and this conclusion would be supported by the previous studies (as written in the revised discussion, line 385–391).

Line 337f: Mechanisms of claustral activity not identified? As mentioned before, there is ample evidence in the literature that provided data for the existence of an ACC-CLA circuit (ACC-CLA: White et al., 2018, Chia et al., 2020) or the specific recruitment of the claustrum during higher cognition (Atlan, 2018, Terem, 2020)

Reply: We apologize our misleading description, and we made it clear what is unknown in the revised manuscript, following the reviewer's suggestion (line 396–397).

Line 350: probably should add a statement that this is in line with previous findings

Reply: Following the reviewer's suggestion, we added the statement with a reference (line 409).

- Line 351: "attention or affect"? What do you mean?

Reply: As the reviewer mentioned, it was ambiguous, and we have replaced the words with internal states in accordance with the references (line 410).

- Line 410: 250 nl injection volume is relatively large, especially for claustral injections - you might have also labelled insular cortex neurons. 1) While insular cortex-ACC projections are rare, they do exist. 2) As the insula is involved in pain processing, optogenetic activation of direct ACC-ins Ctx circuits might contribute to pain processing. Have you checked for any Chronos-labelled insular cortex neurons projecting to the ACC (retroflip ACC- Chronos in claustrum)? Please provide some high-resolution images of the upper insular cortex layers next to the labelled claustrum

Reply: Regarding insular cortex projection, we did not observe labeled neurons in the IC in the anterograde (the revised Supplemental Fig. 3) or retrograde experiments (the revised Fig. 5j, *right panel*). Therefore, we are concluding that the contribution of insular neurons in our observations were

much smaller than that of the CLA.

I also have some editorial comments for the figures:

Figure 1

a. Minor: Panel B: Do you have any examples of a less leaky viral injections?

Reply: As requested by the reviewer, we performed additional experiments using an AAV virus containing pyramidal neuron selective promoter, AAV5-CaMKII α -hM4Di-mCherry, and found that the intensity of terminals of ACC neurons were stronger in the contralateral than the ipsilateral CLA, in accordance with our original data. We have included these data in the revised Supplementary Fig. 1a–d and in the Result section of the revised manuscript (line 85–89).

b. Minor: Panel E-G: due to the spread of retrobeads into the INS cortex, and clearly the endopiriform cortex, the performed cell counts reflect connections to the surrounding areas of the claustrum. The quantification is unlikely to reflect network differences that are limited to the claustrum and hence, should be removed. Also, tiling effect in panel E is disturbing.

Reply: It is true that these data do not completely reflect the claustral connection due to the diffusion of Retrobeads to other areas, but it is one of the results supporting the biased projection of ACC neurons to the contralateral than the ipsilateral CLA, and we could reproduce the same data suggesting the projection preference in the other set of experiment. Therefore, we have kept the data in the revised main figure and added the reproduced data into the summary data, and replace the images with a more specific injection in the revised figure (Fig. 1e–g).

c. Lastly, panel C clearly indicates that ACC projects to both hemispheres, indicating claustral modulation on both sides, indicating that your work only depicts a part of the network. This should be at least be mentioned in the discussion.

Reply: As pointed by the reviewer, in this study we only investigated the role of ACC projection to the contralateral CLA. Therefore, we added the comments about necessity of future works to identify physiological role of the ACC projection to the ipsilateral CLA in the Discussion section of the revised manuscript (line 369–372).

Figure 2

a. Panel A: As the contralateral motor cortex also projects to the claustrum, please provide a figure showing that the recording electrode was in the ACC not the motor cortex... otherwise motor signals could have been interpreted as ACC-> Cla signals

Reply: As requested by the reviewer, we added a representative image showing the electrode track of our recording in the revised figure (Fig. 2b, *left panel*).

b. Panel B: Your optical fiber was close to the strongly labelled external capsule: how did you prevent stimulating passing axons that could be interpreted as claustrum-targeting in ACC electrode recordings (false-positives)?

Reply: As pointed by the reviewer, in the panel, our optical fiber was close to the external capsule and it could cause false positive, but it is difficult for us to completely restrict the light stimulation to the contralateral CLA in our current technique. However, the intensity of GFP-expression in the contralateral CLA is stronger than those of the surrounding areas, and most of axons passing through external capsule would project to CLA because of the selective projections of ACC neurons to the contralateral CLA (see the revised Supplementary Fig. 1e–h). Therefore, we are assuming that the effects of stimulation of external capsule or other adjacent areas were minor.

c. Panel D-F: are those all the recorded units? What's going on top of the plot? There are neurons that seem to fire at a much higher frequency. These neurons don't seem to be affected. Are they INs (spike width) or PN? If these are INs, they shouldn't be included and counted as Cla-projecting (unless you can confirm identity by IHC/marker expression and retrobead/marker expression).

Reply: In our recording method, it is difficult to check the projections of recorded neurons by IHC or Retrobeads after recording, but we judged the recorded neurons as optically tagged or non-tagged from their optical responses, and their latencies and jitters. However, as pointed by the reviewer, some neurons fired in high frequency, and previous reports used neuronal basal firing rate as one of the criteria to classify their recorded cells (Xu et al., *Neuron*, 102(3): 668–682, 2019, and Zhu et al., *Nat Neurosci*, 24(4): 542–553, 2021). Following these previous reports and the reviewer's suggestion, we adapted a new criterion, and removed the unit data whose base line firing rate exceed 20 Hz from the analysis. Even in the new analysis, we could observe the same response preference to the contralateral side in ACC^{→contraCLA} neurons. We have replaced the original data with the new data using the new criteria in the revised main figure (Fig. 2d–g), and added description of the criterion of basal firing rate in the Method section of the revised manuscript (line 605–608).

d. Panel H: What is shown on the top trace (label missing).. how are unit activities indicated in h? Sometimes a large spike is indicated, sometimes not...Please indicate the detection threshold in the top trace to clarify when unit activity was detected and plotted in row 2 ("unit"). What's that rhythmic signal in EMG? Please add short comment non-EMG people into the legend.

Reply: We apologize for missing the label, and it shows the filtered extracellular signals (300-3000 Hz, including neuronal spike wave forms), and we have added the label in the figure. In that figure, we detected two unit activities classified based on their waveforms and antidromic responses using Plexon offline sorter software, where the optically labeled unit exhibit larger amplitude and different

spike kinetics compared to the non-labeled unit. In the original figure, we only showed the optically labeled unit activity, and that of the non-labeled neuron was not shown. However, because it was confusing, we have added non-labeled unit activities and the averaged waveforms of these units in the revised figure (Fig. 2h) and its corresponding legends (line 935–938). The rhythmic signal in EMG is the breathing signal from the diaphragm, and we added the comment in the corresponding legend (line 938).

e. Panel H/I: Unit responses reflect movement on the contralateral body side. Do you also have unit recordings from the ipsilateral side? Is the ACC silent for ipsilateral movements? Basically, how specific does the ACC encode lateralized movements?

Reply: Regarding panel H/I, these data are the traces and summary of the responses to contralateral sensory stimulation with or without following behaviors, suggesting that the magnitude of unit responses to sensory stimulations was modulated by following behavioral consequences. Panel J is the summary data for the responses to ipsilateral sensory stimulation with or without following behavior. We don't assume that the ACC encode lateralized movement because ACC neurons did not exhibit any clear activities during spontaneous movements, and increase of the unit activity by following behaviors was irrelevant to whether the mice responded to the sensory stimulations using a unilateral forepaw or both forepaws. Instead of that, we are assuming that ACC neurons preferentially respond to the lateralized somatosensory inputs because of lateralized projections from the S1 cortex, and the activity levels induced by the sensory inputs is augmented by following behavioral consequences, as ACC neuronal activity represents multiple aspects of pain and associated action planning (see discussion section of the revised manuscript (line 385–391)). However, further studies are needed to investigate the underlying circuit mechanisms of this facilitation. We have included the last part of the above descriptions in the Discussion section of the revised manuscript (line 391–393).

f. As shown in panel D-F, the responses of contralateral-projecting ACC->Cla units reflects reliably contralateral stimulus input (that reaches a ceiling effect quickly), while ipsilateral mechanical stimulations depend on the strength of the stimulus (responses ACC-> Cla units) $0.2g < 0.6g <$ whisker stroking). For H-J. What was the used strength of the mechanical whisker stimulation? For ipsilateral stimuli (j), was the chosen stimuli to weak (0.16g) to elicit a response in the first place (see ipsilateral stimuli at 0.16g in panel d).

Reply: In the original and revised Fig. 2F, we used whisker stroking stimulation using a closed forceps as written in the Method section (line 608–611). In the Fig 2h–j, we used whisker stroking stimulations to ipsilateral and contralateral sides, and only in the contralateral stimulation experiment, the following behavioral consequences related with the magnitude of ACC neuronal responses to stimulation.

Figure 3:

a. Panel D: Why are there multiple EPSCs? Explanation for multiple EPSCs is missing

Reply: Regarding the former comment from the reviewer 1, we moved these recording data to revised Supplementary Fig. 2, but as pointed by the reviewer, we added the explanation for the EPSCs in the legend in the revised supplementary information (line 34–35 in Supplementary information.).

b. Panel H: Looks like some ipsilateral axons also target contra and ipsilateral striatum and ventral brain areas ...are those blood vessel artifacts or actually axons?

Reply: As we have showed in Supplementary Fig. 3, those signals seem to be actual axons.

c. Panels J-P: You are showing two representative neurons, not the morphological features. A graphical representation of connectivity percentage (EPSCs) and percentage of AP triggered would great. Also, because PNs might have higher Rms than FS INs, a lower EPSC might result in comparable or higher depolarizations than observed in FS Ins. What is the average EPSP amplitude for INs vs PNs? And lastly, as you observed a nearly significant difference of the paired pulse ratio, how does these depolarization differences translate for a train of stimuli (10Hz, 20Hz or even higher)? It seems like the second EPSC for PNs is higher than for INs... Could that imply a change in the preferred population of recruited neurons (initial recruitment INs, later PNs)? Any additional info's could potentially help to settle the ongoing discussion of claustral effects being either excitatory or inhibitory.

Reply: As pointed by the reviewer, we corrected the description in the revised Legend (line 950). We also added description of connectivity rate and rate of AP triggered neurons in the revised main figure (Fig. 3f, g, i, j, m, n, q, r). As requested by the reviewer, we performed additional analysis of EPSP amplitude of INs and PNs, and found that EPSP amplitude of FS-INs was also higher than that of PNs. Regarding the latter comment and the reviewer's other comment, we performed additional *in vivo* LFP recording using higher frequency light stimulation of the terminals of ACC neurons in the contralateral CLA. In that data, the EEG delta power was increased by all frequency stimulation, which is associated with activation of fast-spiking interneurons. Interestingly, amplitude of the facilitation at 20 Hz stimulation was significantly lower than those induced by the other frequencies. This phenomenon could be partly explained by the paired pulse depression in the CLA to ACC FS-INs synapse, by which the amplitude of EPSP could become insufficient to recruit the AP firings in ACC FS-INs due to the faster decay. We have included these data in the revised figure (Fig. 3v) and the revised supplementary information (Supplementary Fig. 4), and added a part of descriptions in the Result section of the revised manuscript (line 176–180, 197–204).

Figure 4:

a. Panel B: color scale is missing; also...after each stimulation you can observe an increase in the power range between 10 and 20Hz...is that significant?

Reply: We apologize for missing the color scale and now added the scale. Regarding the higher frequency power between 10 and 20 Hz, there was significant difference (Pre, 0.55 ± 0.09 ; Post, 1.24 ± 0.18 , two-tailed paired t-test, $p < 0.001$). These data suggested that activation of ACC \rightarrow contraCLA neurons facilitates not only the EEG delta power but also the EEG power in higher frequency range. A previous report has suggested that β -frequency oscillation (14-30 Hz) is also important for decision making via cortical inhibitory circuit (Dubey et al, Neuron, 111, 3321-3334, 2023), and it is possible that CLA neurons participate in these higher order decision-making processes by recruiting cortical inhibitory circuit, as our study and other studies have shown. These results are interesting, but it will take so long time to elucidate roles of the higher frequency EEG modulation via the CLA that we have decided to investigate their roles in the next study.

b. Panel E and figure legend: The effect on PNs seem to be bidirectional: first an inhibition, followed by a period of increased activity around 200ms after the stimulus.

Reply: As pointed by the reviewer, we have added the mention about rebound firing in the corresponding legend (line 981–983).

c. Panel G-I and J: compare the peak frequencies. In general, the frequency peak in the average PTSH in the bottom of G-I is higher than the values in J. Why? Also, how was J quantified? Seems like peak amplitude wise, panel G responses showed no differences in the PN firing rate but are indicated as significantly lower in J? The panels E and G-J don't support each other and the observed inhibitory responses in G-I only start to be effective after an initial increase of AP firing. How do you explain those effects (fast unit inhibition in E, population activation in G-I even in the presence of CLA \diamond ACC activation; late inhibitory effects)

Reply: In the original Fig 4J, we calculated the subtractions of averaged firing frequency (for 2 s) before the stimuli from that after the stimuli as we have written in the method section (line 611–613). As pointed by the reviewer, the effects of light stimulation seemed to be weak in the original data. In these original experiments, we used 20 Hz stimulation protocol to activate the ACC \rightarrow contraCLA neuronal terminals to evaluate its effects on the sensory responses elicited in contralateral ACC units. However, in the former SW analysis using different frequency stimulations regarding the previous comments, 20 Hz stimulation only induced weaker EEG delta power increase compared to the other lower frequency stimulations (the revised Supplementary Fig. 4). Therefore, we assumed that because we used too high frequency stimulation to recruit inhibitory circuits in the contralateral ACC efficiently, there were not obvious differences in the peak frequencies in the original data (the original Fig. 4g–j). Then, we performed additional experiments using a 10 Hz optical stimulation protocol, and the light

stimulation significantly reduced the peak frequencies of the responses and the averaged firing changes compared to the responses without light stimulation. We have replaced the original data with the 10 Hz light stimulation data in the revised figure (Fig. 4g–j) and changed the corresponding text in the Result section and Legend section of the revised manuscript (line 222, line 986–991).

d. Panel N: The chemogenetic activation of neurons appear to be constant. That is troublesome as you lose the ability to decode time-sensitive information that are probably critical under real-life conditions (acute pain events).

Reply: Following the reviewer’s suggestion, we performed additional optogenetic experiments using a 10 Hz stimulation protocol, and confirmed the original hypothesis that ACC^{→contraCLA} neurons inhibit the ipsilateral sensory responses. Furthermore, we also performed an optogenetic anterograde experiment, and we observed the similar inhibition of sensory responses, suggesting that the behavioral changes induced by activation of ACC^{→contraCLA} neurons were mediated by the contralateral CLA. We have included these data in the revised figure (Fig. 4k–q) and in the Result section of the revised manuscript (line 226–239).

Figure 5:

a. Panel 5d: statistical test results for hind paw results were shown in main text, the attending behaviour was not shown (is just one more line)... why?

Reply: Due to the limitation of the number of words of the main text, we decided to show the behavioral statistical results only in the figure (the revised Fig. 5b, c), but not in the text of the revised manuscript.

b. Panel 5j: AAV9 labels anterograde and retrograde cell populations, chemogenetic effects are not region-specific. Do you have brain slide images for mCherry expression in the claustrum to confirm that you are not inhibiting the claustrum directly.

Reply: As pointed by the reviewer, we have included an image showing mCherry expression in the contralateral CLA in the revised figure (Fig 5j, *right panel*) and corresponding text in the Legend section of the revised manuscript (line 1029–1031), confirming that we did not manipulate the CLA directly.

c. Compare 5f and 5j: both panels show retrograde labelling of Cla projecting ACC neurons. Why is it that in panel 5f, the labelled neurons are more superficial, while in 5j most claustral projecting neuron are in deep layers?

Reply: We did not observe any clear difference between these expression patterns between these experiments, but we apologized that we used the misleading image in the original manuscript, and we

have replaced the GCaMP6s image with the other image in the revised figure (Fig. 5f).

Figure 6:

a) In general, after establishing the ACC-> contralateral CLA-> ipsi ACC-> Ipsi ACC loop, it would be good to graphically represent this loop including the postulated effects of mechanical sensation and ACC activity.

Reply: As requested by the reviewer, we have included the graphical abstract of this article in the revised figure (Fig. 7) and the corresponding legend (line 1068–1070).

REVIEWERS' COMMENTS:

Reviewer #1 (Remarks to the Author):

The authors have performed several additional experiments to support their claims. The paper is very dense, which is not necessarily a bad thing. I have many questions regarding the results, but none that preclude publication at this time. I congratulate them on a very interesting paper.

Reviewer #2 (Remarks to the Author):

The authors have conscientiously addressed all of the points that I raised previously. They have also clarified uncertainties, improved figures, and added results from several additional experiments to the manuscript.

With these extensive revisions, the manuscript is improved in clarity and is more convincing. I support the publication in the journal but have 2 very minor editorial points for the authors to consider:

1) given all the additional experiments that the authors have done, I recommend that the abstract be updated to include the optogenetic experimental results, rather than the supplementary chemogenetic results.

2) The authors (and many other authors in the field) erroneously refer to their electrophysiological data with constructions similar to this: "Recordings of CLA→ipsiACC neurons". This is poor English, because the neurons themselves are not being recorded; instead, it is their responses that are being recorded. I recommend that the authors improve their word usage in such situations, which occur throughout the ms.

re: comments of Reviewer 1

The authors have performed several additional experiments to support their claims. The paper is very dense, which is not necessarily a bad thing. I have many questions regarding the results, but none that preclude publication at this time. I congratulate them on a very interesting paper.

Reply: We are pleased that Reviewer 1 finds that our study “is a very interesting paper”. We thank again him/her for the time and the previous comments to improve our study.

re: comments of Reviewer 1

The authors have conscientiously addressed all of the points that I raised previously. They have also clarified uncertainties, improved figures, and added results from several additional experiments to the manuscript. With these extensive revisions, the manuscript is improved in clarity and is more convincing. I support the publication in the journal but have 2 very minor editorial points for the authors to consider:

Reply: We are pleased that Reviewer 1 finds that our revised manuscript “is improved in clarity and is more convincing”. We appreciate his/her support for the publication of our study. We have modified several points following the reviewer’s suggestion in the revised manuscript.

1) given all the additional experiments that the authors have done, I recommend that the abstract be updated to include the optogenetic experimental results, rather than the supplementary chemogenetic results.

Reply: Following reviewer’s suggestions, we have added the optogenetic behavioral experimental results in the abstract in the revised manuscript (line 36–39).

2) The authors (and many other authors in the field) erroneously refer to their electrophysiological data with constructions similar to this: "Recordings of CLA→ipsiACC neurons". This is poor English, because the neurons themselves are not being recorded; instead, it is their responses that are being recorded. I recommend that the authors improve their word usage in such situations, which occur throughout the ms.

Reply: We apologized our mistakes, and we checked and corrected such expressions throughout the manuscript (line 141).

Additionally, we apologize that we had several mistakes related to the reviewer’s previous comments in our previous revised manuscript. Regarding Figure 2b left panel, we mistakenly exhibited the flipped image. Furthermore, regarding Figure 2h, the rhythmic signals in EMG are pulsation signals, but not breathing signals from the diaphragm. We have corrected these points in the revised Figure (Fig. 2b) and the legend section of the revised manuscript (line 946).